# Seasonal dynamics of stem N$_2$O exchange follow the physiological activity of boreal trees

Katerina Machacova [1]*, Elisa Vainio [2,3], Otmar Urban [1] & Mari Pihlatie [2,3,4]

The role of trees in the nitrous oxide (N$_2$O) balance of boreal forests has been neglected despite evidence suggesting their substantial contribution. We measured seasonal changes in N$_2$O fluxes from soil and stems of boreal trees in Finland, showing clear seasonality in stem N$_2$O flux following tree physiological activity, particularly processes of CO$_2$ uptake and release. Stem N$_2$O emissions peak during the vegetation season, decrease rapidly in October, and remain low but significant to the annual totals during winter dormancy. Trees growing on dry soils even turn to consumption of N$_2$O from the atmosphere during dormancy, thereby reducing their overall N$_2$O emissions. At an annual scale, pine, spruce and birch are net N$_2$O sources, with spruce being the strongest emitter. Boreal trees thus markedly contribute to the seasonal dynamics of ecosystem N$_2$O exchange, and their species-specific contribution should be included into forest emission inventories.

[1] Global Change Research Institute of the Czech Academy of Sciences, Belidla 4a, CZ-60300 Brno, Czech Republic. [2] Environmental Soil Science, Department of Agricultural Sciences, University of Helsinki, P.O.Box 56FI-00014 Helsinki, Finland. [3] Institute for Atmospheric and Earth System Research/Forest Sciences, University of Helsinki, P.O.Box 27FI-00014 Helsinki, Finland. [4] Viikki Plant Science Centre (ViPS), University of Helsinki, P.O.Box 56FI-00014 Helsinki, Finland. *email: machacova.k@czechglobe.cz

With an area of about 1370 million ha, boreal forests comprise one-third of the global forested area[1,2]. Boreal forests cover much of the uppermost Northern Hemisphere (Canada, Russia, and Scandinavia). In Finland, they consist of a mosaic of different forest types, ranging from upland forests on mainly dry mineral soils with scattered small-scale peatland areas to peatlands with tree cover[3]. Boreal forests are considered a natural source of nitrous oxide ($N_2O$), an important greenhouse gas with global warming potential of 298 over 100 years[4]. The net $N_2O$ emissions are estimated to be 0.38 kg ha$^{-1}$ yr$^{-1}$ based on predominant $N_2O$ production in the soil[2]. Even though $N_2O$ fluxes in boreal forests are lower compared to those of temperate and tropical forests (1.57 and 4.76 kg ha$^{-1}$ yr$^{-1}$)[2], boreal forests play an important role in global $N_2O$ inventories due to their large area.

$N_2O$ is naturally produced within soils in a wide range of nitrogen (N) turnover processes, including mainly nitrification and denitrification processes, which can be closely interconnected. The denitrification processes are the only processes known to reduce $N_2O$ to $N_2$[5–7]. Generally, plants can contribute to ecosystem $N_2O$ exchange by taking up $N_2O$ from soil water and transporting it into the atmosphere through the transpiration stream[8,9], by producing $N_2O$ directly in plant tissues[10], by consuming $N_2O$ from the atmosphere by a non-specified mechanism[11], and by altering the N turnover processes in adjacent soil[12].

Recent research has revealed that not only herbaceous plants but also some tree species can be significant $N_2O$ sources[13,14]. High $N_2O$ emissions have been detected, however, only in the laboratory from seedlings grown under conditions of artificially increased $N_2O$ concentrations in soils[8,9,15,16]. Ecologically relevant studies with mature trees growing in natural field conditions are rare and have revealed only low $N_2O$ emissions[13,14,17] or even consistent $N_2O$ consumption from the atmosphere[11]. Because these studies were often conducted without a full series of supporting environmental and physiological measurements, interpretation of the observed $N_2O$ fluxes is therefore limited.

Moreover, seasonal measurements of $N_2O$ fluxes are lacking, and particularly during the dormant season. Therefore, no information exists about possible seasonality of tree $N_2O$ fluxes. Another shortcoming is that calculation of annual $N_2O$ fluxes to date has been based solely on the results of short measuring periods during the vegetation season. It is well known that physiological activity of boreal trees is strongly reduced during winter, including photosynthetic $CO_2$ assimilation, transpiration, and sap flow[18,19]. Transport of $N_2O$ in the transpiration stream has been suggested as one mechanism for $N_2O$ emissions from tree stems and canopies[8,20]. Also, possible $N_2O$ production in plant tissues during nitrate assimilation[10,12,21–23] seems to be closely connected to photosynthesis, which is a process requiring sufficient light intensity, temperature, water, and mineral supplies. One must therefore conclude that lack of comprehensive understanding of the physiological and environmental drivers of variability in tree $N_2O$ flux leads to poor understanding of the $N_2O$ dynamics. The determination of $N_2O$ exchange rates of common tree species and their dynamics is of high importance for correct estimation of forest $N_2O$ budgets and therefore of global greenhouse gas flux inventories.

Accordingly, we hypothesised that tree-stem $N_2O$ fluxes have substantial seasonal dynamics in boreal forest and that the stem $N_2O$ fluxes are related to tree physiological activity as driven by such environmental variables as temperature, light intensity, and water availability. We quantified $N_2O$ fluxes of three dominant tree species—coniferous Scots pine (*Pinus sylvestris* L.) and Norway spruce (*Picea abies* L. Karst.), and broadleaved downy and silver birch (*Betula pubescens* Ehrh., *B. pendula* Roth.)—grown at plots naturally differing in soil volumetric water content

(VWC). Seasonal $N_2O$ fluxes were measured on mature trees in southern Finland together with forest floor $N_2O$ and $CO_2$ fluxes, numerous physiological parameters, and a range of environmental parameters describing meteorological, soil and atmospheric conditions. Stem $CO_2$ effluxes as well as ecosystem gross primary productivity (GPP) and evapotranspiration were considered as indicators of physiological activity of the trees. Such a comprehensive experimental setup with multiple environmental variables measured together with $N_2O$ fluxes enabled us to investigate whether these tree species exchange $N_2O$ with the atmosphere, whether tree $N_2O$ fluxes have seasonal dynamics, whether trees exchange $N_2O$ during winter dormancy, whether $N_2O$ fluxes are related to environmental and physiological parameters, and how trees contribute to the net ecosystem $N_2O$ exchange. We discovered a strong seasonality in stem $N_2O$ exchange, which tightly relates to the physiological activity of the trees, in particular to $CO_2$ uptake and release. We show that boreal trees exchange small amounts of $N_2O$ even during the dormant winter season. This unique dataset of whole-year $N_2O$ stem fluxes revealed boreal trees as net annual sources of $N_2O$, with spruce being the strongest emitter. Our study thus provides a comprehensive overview of tree $N_2O$ flux dynamics and their environmental drivers in the soil—tree stem—atmosphere continuum.

## Results and discussion

**Stem $N_2O$ flux seasonality relates to physiological activity.** All boreal trees studied showed substantial seasonality in their $N_2O$ exchange. Previous rare studies have focused only on short periods of the vegetation season, excluding measurements in the dormant winter season. High and constant stem $N_2O$ emissions were observed from all the tree species studied during spring and summer months (April–September), independently of soil VWC. This was followed by a decrease from October onwards (Fig. 1). Tree fluxes remained low in winter and increased again in March. Stem $CO_2$ effluxes revealed similar seasonality as did $N_2O$ fluxes (Supplementary Fig. 1). Seasonal dynamics of accompanying environmental parameters are presented in Fig. 2. A strong positive relationship between stem $N_2O$ and $CO_2$ fluxes was detected ($\rho = 0.714$–$0.745$, depending on tree species; $p < 0.001$; Spearman's rank correlation), and reduced flux rates were observed in the dormant season (Fig. 3). The positive relationship was further supported by a partial least squares (PLS) path analysis (Fig. 4, Supplementary Figs. 2, 3). Based on the results of the PLS path and correlation analyses, the second main driver of the stem $N_2O$ fluxes is the GPP of the forest ($\rho = 0.543$–$0.660$, $p < 0.001$; Fig. 4, Supplementary Figs. 2, 3). For birch and pine, the stem $CO_2$ efflux and GPP together explained 44% and 37%, respectively, of the variance in the stem $N_2O$ fluxes (Supplementary Figs. 2, 3).

From the available environmental variables, the stem $N_2O$ fluxes of all the studied tree species correlated positively ($p < 0.001$) with air temperature ($\rho = 0.559$–$0.645$), as well as intensities of photosynthetically active radiation (PAR) ($\rho = 0.513$–$0.637$) and ultraviolet (UV) radiation ($\rho = 0.541$–$0.694$). We also found strong positive correlation between the stem $N_2O$ fluxes and ecosystem evapotranspiration ($\rho = 0.536$–$0.688$, $p < 0.001$). That relationship was not proven by the PLS path analysis, however.

The relatively strong relationship between the stem $N_2O$ fluxes and those variables reflecting the physiological activity of the trees and ecosystem as a whole (stem $CO_2$ efflux, GPP, evapotranspiration) suggests that the stem $N_2O$ fluxes do not constitute merely a passive process based on $N_2O$ concentration gradients in the soil–stem–atmosphere continuum. The possible coupling

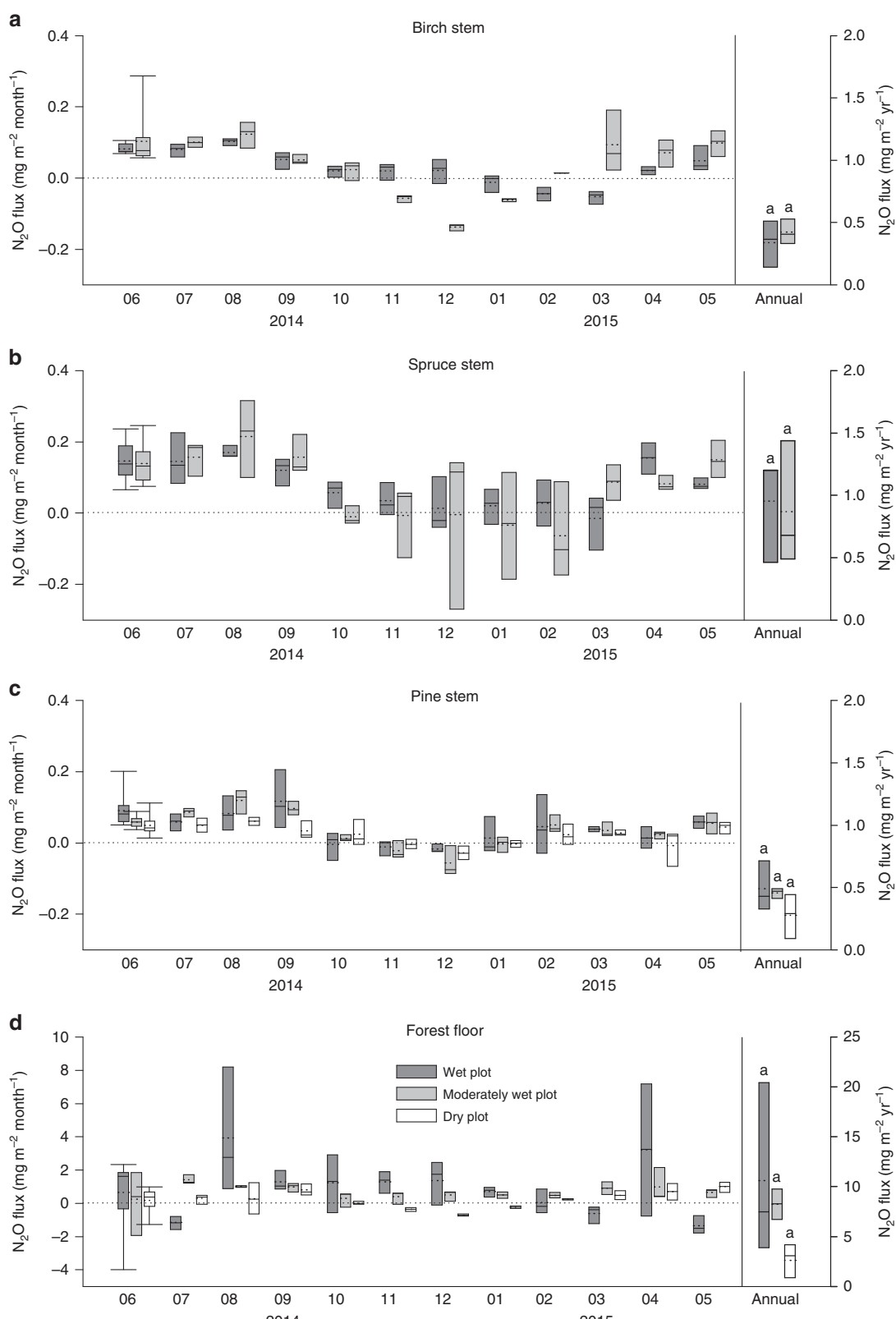

**Fig. 1** Stem and forest floor $N_2O$ fluxes. Seasonal courses of monthly $N_2O$ fluxes (mg m$^{-2}$ month$^{-1}$) and total annual $N_2O$ fluxes (mg m$^{-2}$ yr$^{-1}$) from stems of birch (**a**), spruce (**b**), and pine (**c**), and from forest floor (**d**) measured from June 2014 to May 2015. Positive fluxes indicate $N_2O$ emission, negative fluxes $N_2O$ uptake. The solid line within each box marks the median value, broken line the mean, box boundaries the 25th and 75th percentiles, and whiskers the 10th and 90th percentiles. Statistically significant differences among annual fluxes at $p < 0.05$ are indicated by different letters above bars. Mean annual volumetric water contents (± standard error) of the plots were as follow: wet plot, 0.81 ± 0.02 m$^3$ m$^{-3}$; moderately wet plot, 0.40 ± 0.02 m$^3$ m$^{-3}$; and dry plot, 0.21 ± 0.01 m$^3$ m$^{-3}$. The dry plot did not have spruce or birch trees. Stem fluxes were measured from three trees per species at each plot ($n = 3$). Forest floor fluxes were measured at three positions at the wet and moderately wet plots ($n = 3$) and at six positions at the dry plot ($n = 6$). Annual fluxes were calculated as the sums of 12 monthly fluxes

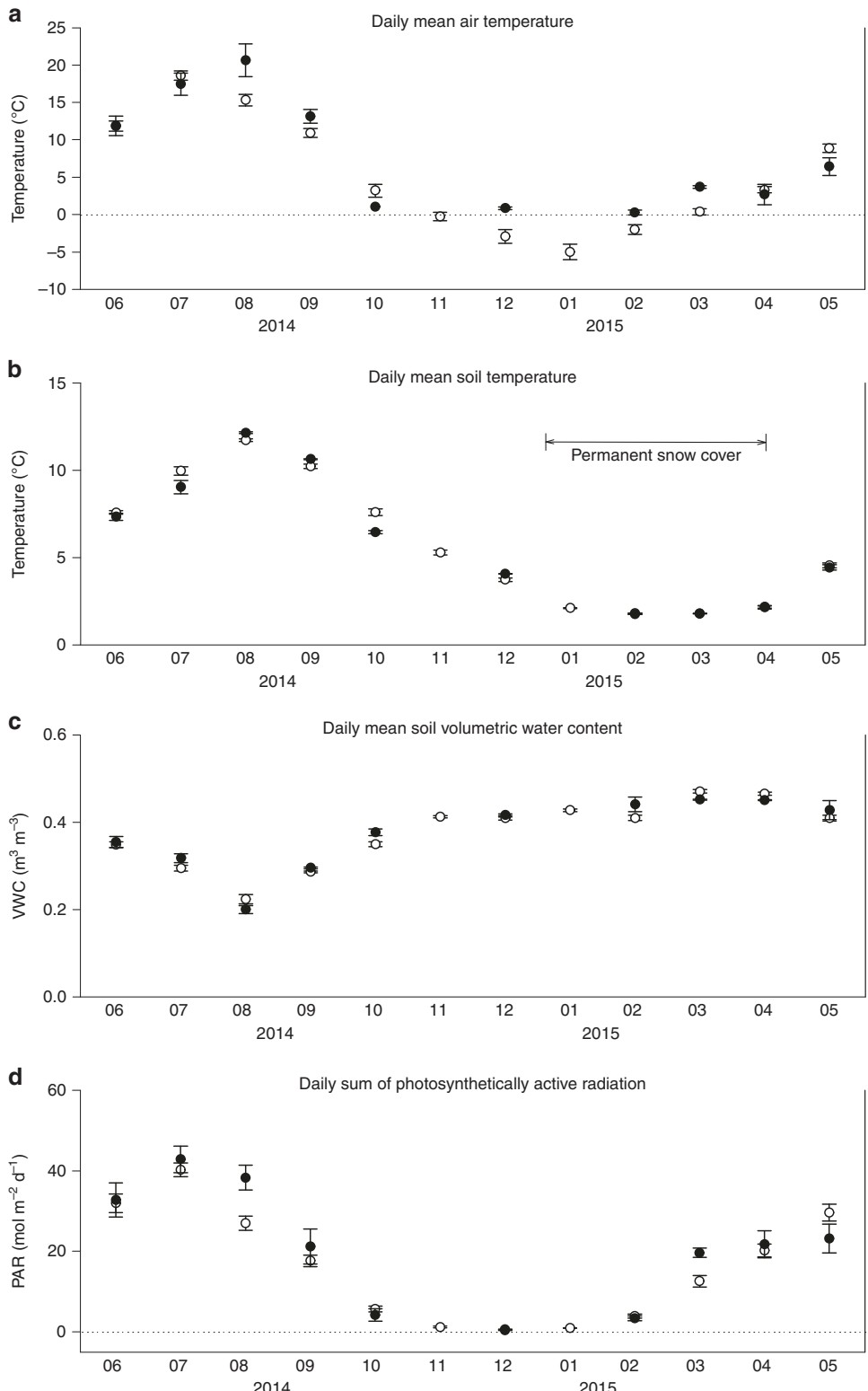

**Fig. 2** Seasonal courses of basic environmental variables. The variables were measured at the SMEAR II station from June 2014 to May 2015: (**a**) Daily mean air temperature within the forest stand at 8 m height; (**b**) daily mean soil temperature and (**c**) soil volumetric water content (VWC), both in C1-horizon (38–60 cm in depth); and (**d**) daily sum of photosynthetically active radiation (PAR). The open circles represent monthly means (± standard error), and the black circles indicate monthly means ( ± standard error) calculated for the flux measurement days only. The period of continuous snow cover (from mid-December 2014 to early April 2015) is indicated in Fig. 2b

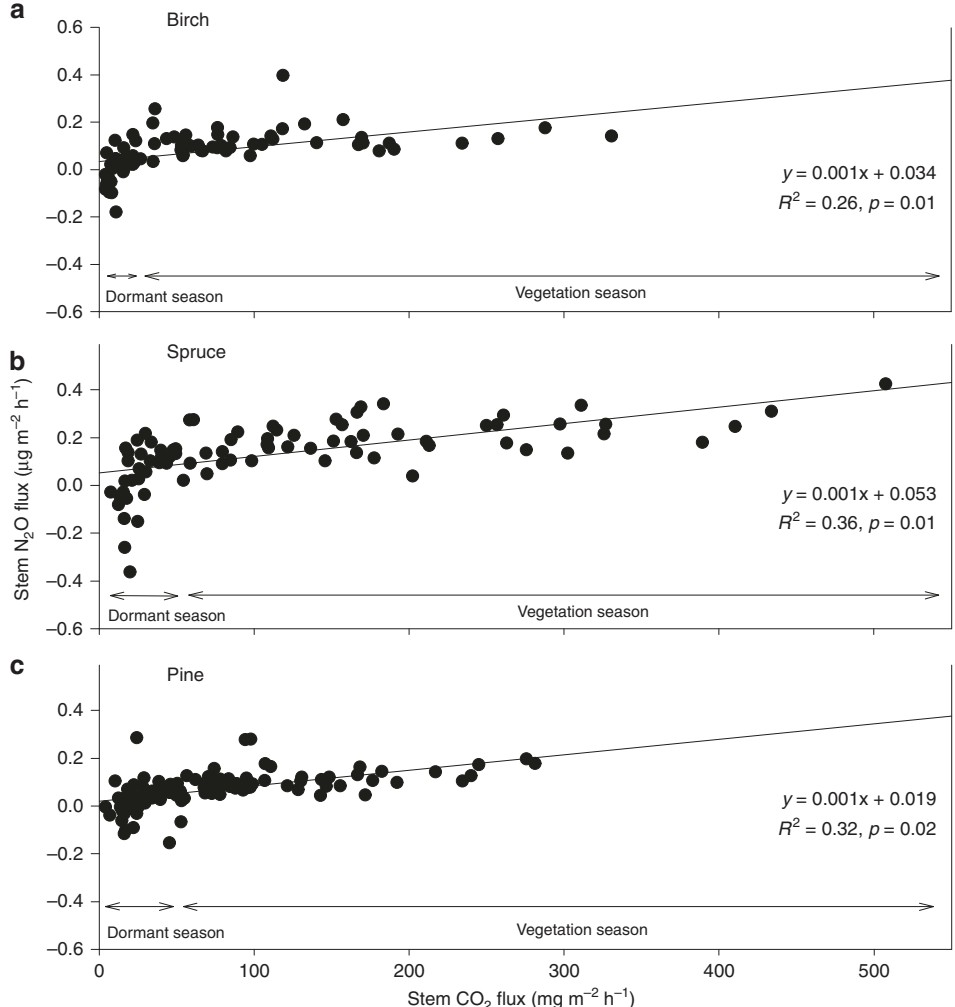

**Fig. 3** Relations between $N_2O$ and $CO_2$ stem fluxes. $N_2O$ versus $CO_2$ fluxes in stems of birch (**a**), spruce (**b**), and pine (**c**) measured from June 2014 to May 2015. Data for dormant (October–February) and vegetation (March–September) seasons are indicated. The stem fluxes were measured from three trees per species at three studied plots characterised by mean annual volumetric water content as wet ($0.81 \pm 0.02$ m$^3$ m$^{-3}$; mean ± standard error), moderately wet ($0.40 \pm 0.02$ m$^3$ m$^{-3}$), and dry ($0.21 \pm 0.01$ m$^3$ m$^{-3}$). The dry plot did not have spruce or birch trees. All fluxes are expressed per m$^2$ of stem surface area. Positive flux values indicate gas emission, negative values gas uptake

between $N_2O$ and $CO_2$ fluxes was early detected in species belonging to cryptogamic covers[11,24] and Spermatophyta[11,25]. This assumption is based on the finding of constant $N_2O$:$CO_2$ emission ratio under a wide range of controlled environmental conditions[24,25]. However, here we show for the first time a tight linear correlation between $N_2O$ and $CO_2$ fluxes even in adult trees during the whole vegetation period (Fig. 3) supporting thus the hypothesis of physiologically dependent $N_2O$ exchange in tree stems.

The strong positive correlation with evapotranspiration supports our hypothesis that $N_2O$ is taken up from the soil by roots, then transported into the above-ground tree tissues in xylem via the transpiration stream. This hypothesis is supported also by the good solubility of $N_2O$ in water[26] and the demonstrated ability of plants lacking the aerenchyma system[8,9,13] to transport $N_2O$ from the soil and emit it through the stem. To our knowledge, this is the first study showing that $N_2O$ exchange by mature boreal tree stems is closely connected to the physiological activity of trees and ecosystem, particularly to processes of carbon release and uptake, including stem $CO_2$ efflux and GPP. Future experimental studies are needed, however, to confirm that transpiration rate drives $N_2O$ emissions from tree stems.

Forest floor $N_2O$ flux together with the stem $CO_2$ efflux and GPP explained 45% of the stem $N_2O$ flux variance in spruce (Fig. 4). Spruce was the only tree species manifesting a weak relationship between stem and forest floor $N_2O$ fluxes ($\rho = 0.353$, $p < 0.001$) (Fig. 4). Similarly, a positive but weak correlation between stem and forest floor $N_2O$ fluxes was found earlier in pine trees grown in the same forest[13]. Lack of strong correlations indicates a partial decoupling of stem and forest floor $N_2O$ fluxes. Generally, net fluxes at the tree stem/soil–atmosphere interface reflect a balance between processes of production, consumption, and transport of $N_2O$ within trees and soil from the sites of production to the sites of release[27]. Accordingly, substantial variation in root depth among tree species can contribute to observed species specificity in $N_2O$ fluxes. The net forest floor $N_2O$ flux does not necessarily reflect the $N_2O$ concentration or production/consumption in the rooting zone[11], because the plants can directly alter soil microbiological N turnover processes[28–31] by modifying the quantity and quality of soil organic matter, nutrient availability, and soil pH[29,32]. This includes release of exudates[33] and radial oxygen loss[34] from the roots, which are again closely connected to such tree physiological processes as photosynthesis[35]. Furthermore, leaf and root litter

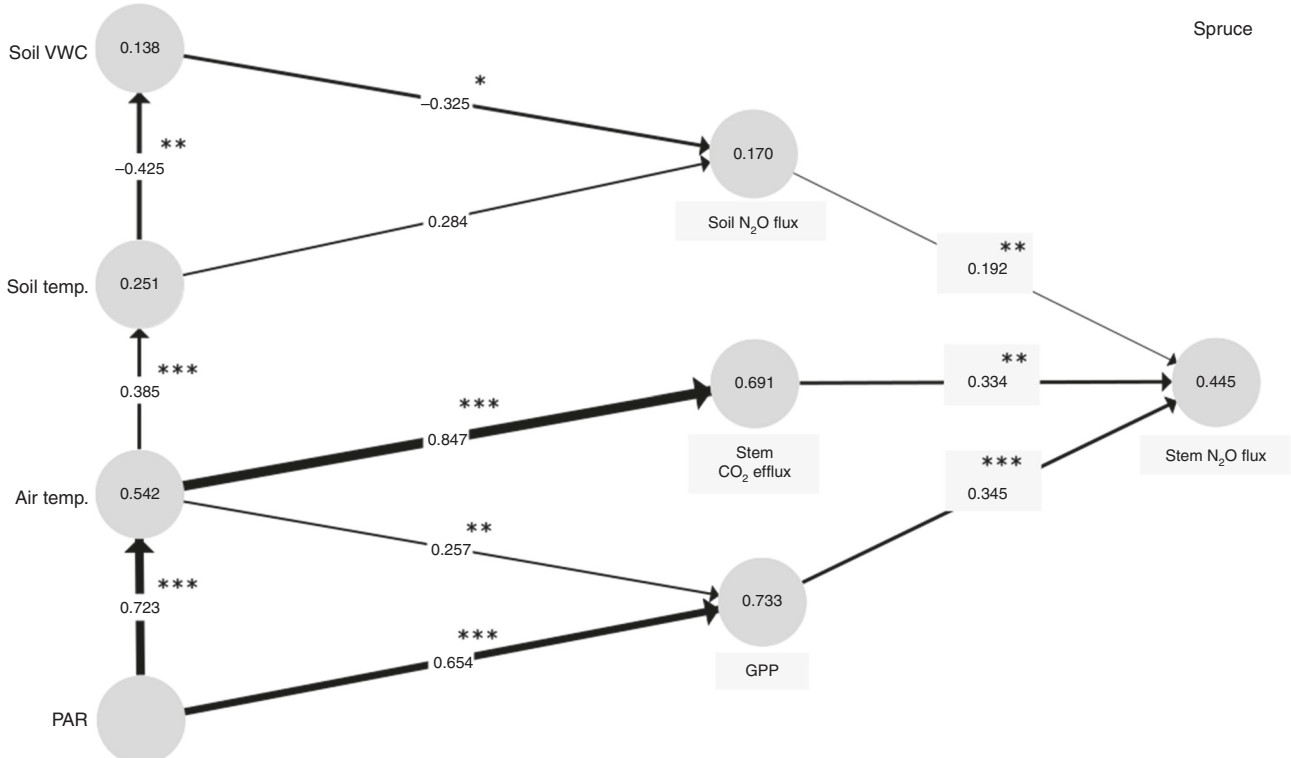

**Fig. 4** Prediction of $N_2O$ fluxes in spruce stem. Path diagram, created on the base of partial least squares path modelling, describes relationships among stem $N_2O$ fluxes and most predictive environmental, physiological, and ecosystem variables (drivers of $N_2O$ fluxes) for 2014–2015. Values in circles report coefficients of determination ($R^2$). Values included in arrows mark the path coefficients, whose significance levels are expressed as follows: *$p < 0.05$, **$p < 0.01$, ***$p < 0.001$. Soil volumetric water content (Soil VWC), photosynthetically active radiation (PAR), gross primary production (GPP). Soil $N_2O$ flux expresses forest floor $N_2O$ flux

quality and soil water uptake by trees, both of which are specific to tree species, can substantially affect the N cycling processes in soils[28]. Production of $N_2O$ in soil is further directly and indirectly influenced by an activity of mycorrhizal fungi via the modulation of denitrification processes and physio-chemical soil properties regulating $N_2O$ turnover like carbon, nitrogen, and water availability, as well as soil aeration and promotion of soil aggregation, respectively[36–39]. Mycorrhizal fungi itself seem to possess also the ability for denitrification and might be therefore important sources of $N_2O$[36,38]. Due to these strong rhizospheric effects of plants and mycorrhizal fungi on soil N turnover processes, the ratio between $N_2O$ production and consumption in the soil might also be highly variable. Hence, the availability of $N_2O$ in the rhizosphere affects the $N_2O$ uptake by tree roots and subsequent $N_2O$ emissions from tree surfaces into the atmosphere, and that might not be reflected in the overall net $N_2O$ exchange at the soil surface.

In addition to that of soil origin, $N_2O$ emitted by trees can also be formed directly in the tree tissues. The direct $N_2O$ production in plants is proposed to originate from microorganisms living in association with the plants[36], as described earlier with mycorrhizal associations, or from $N_2O$ produced via photo-assimilation of $NO_3^-$ in photosynthetically active tree tissues[10,21,22], or via a newly detected biotic pathway with mechanisms different from known microbial or chemical processes[25], or via an abiotic UV-dependent process on leaf surfaces[40]. The plant's own $N_2O$ production process seems to be light dependent, requiring energy from primary photosynthetic reactions[10,12,23]. The mechanisms and processes behind radiation induced $N_2O$ emissions are still poorly understood, however, and especially with respect to mature trees. Moreover, the possible contributions of the various

$N_2O$ production processes in plants to the net $N_2O$ fluxes at the tree–atmosphere continuum are largely unknown. To the best of our knowledge, only Machacova et al.[13] have reported $N_2O$ emissions from leaves of mature trees and showed that the leaf emissions might considerably exceed the emissions from the stems and could therefore constitute an additional source of $N_2O$ in forest ecosystems[13].

In conclusion, the $N_2O$ emission rates from tree stems show clear seasonal dynamics with the highest emissions detected during summer months when also air temperature, PAR and UV intensities are the highest. The seasonal changes in $N_2O$ emission closely relate to the physiological activity of trees associated with $CO_2$ exchange as demonstrated by a tight linear correlation between $N_2O$ and $CO_2$ fluxes.

**Boreal trees exchange $N_2O$ even during dormant season**. Based on the seasonal changes in stem $CO_2$ efflux (Supplementary Fig. 1a–c), the period from October to February was identified as a dormant season. In addition to the vegetation-season $N_2O$ emissions, our study revealed that all the studied tree species can emit $N_2O$ even during the dormant season, and particularly on the plot characterised by high soil VWC (i.e. the wet plot). At this plot, the stem $N_2O$ emissions over the dormant season contributed from 2% (birch) to as much as 16% (spruce) to the annual $N_2O$ emissions (Fig. 5a–c).

The small but detectable winter $N_2O$ fluxes of the tree stems were accompanied by low but consistent $CO_2$ emissions from the stems (Supplementary Fig. 4a–c), thereby reflecting the rate of maintenance respiration during the dormant period[41]. This is supported by the fact that air temperatures were generally mild

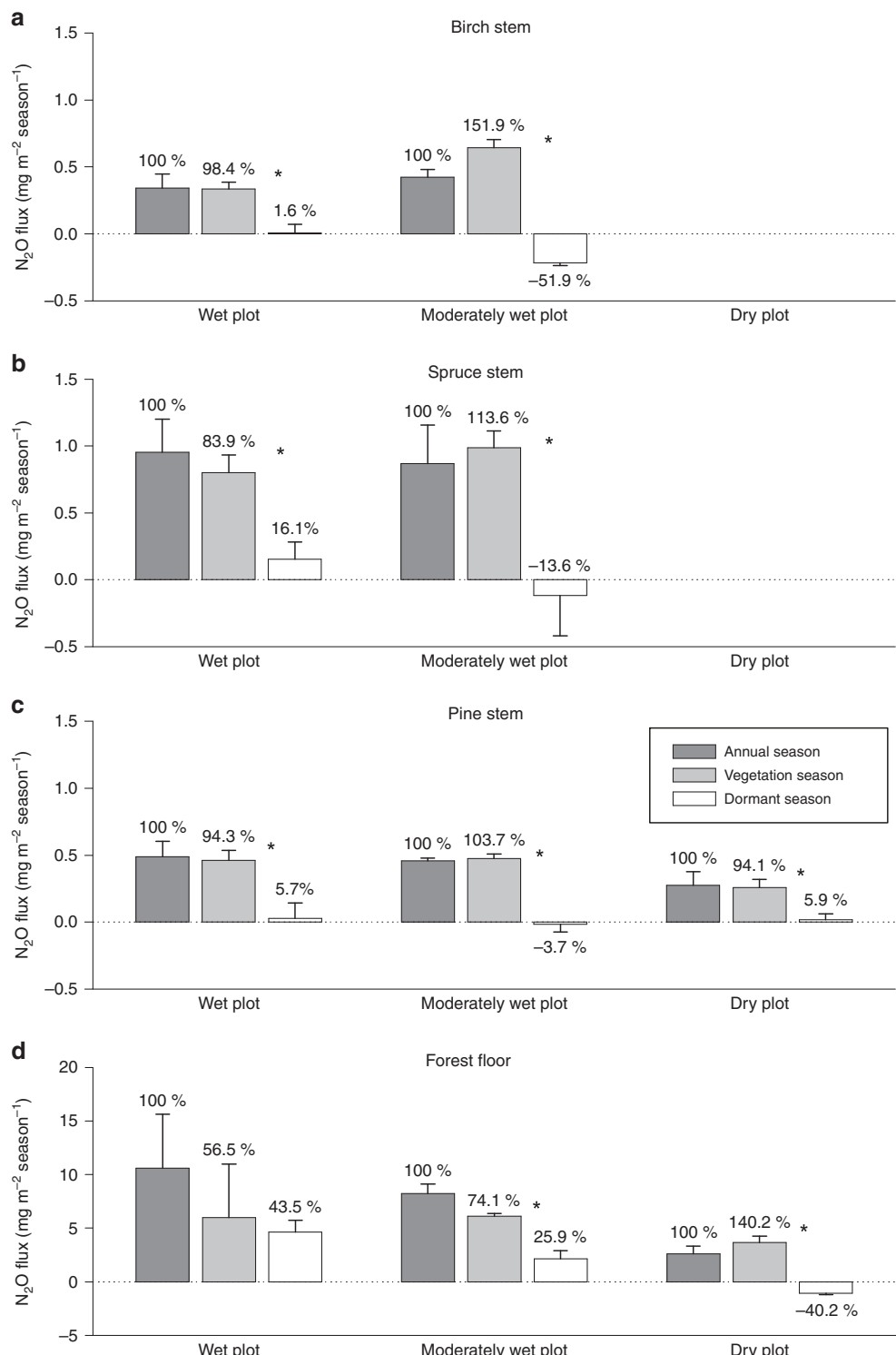

**Fig. 5** Seasonal $N_2O$ fluxes in tree stems and forest floor. $N_2O$ fluxes in stems of birch (**a**), spruce (**b**), and pine (**c**), and in forest floor (**d**) are presented at annual scale (black columns), for vegetation season (March–September, grey columns), and for dormant season (October–February, white columns). The fluxes (means ± standard error) are sums of $N_2O$ exchanged over one year, vegetation season, or dormant season, respectively, and expressed per $m^2$ of stem or soil surface area. Positive flux values indicate $N_2O$ emission, negative values $N_2O$ uptake. Mean annual volumetric water contents ( ± standard error) of the plots were as follow: wet plot, 0.81 ± 0.02 $m^3 m^{-3}$; moderately wet plot, 0.40 ± 0.02 $m^3 m^{-3}$; and dry plot, 0.21 ± 0.01 $m^3 m^{-3}$. The dry plot did not have spruce or birch trees. Stem fluxes were measured from three trees per species at each plot ($n = 3$). Forest floor fluxes were measured at three positions at the wet and moderately wet plots ($n = 3$) and at six positions at the dry plot ($n = 6$). Statistically significant differences between fluxes over vegetation and dormant season at $p < 0.05$ are indicated by asterisks. The percentage contributions of fluxes over the vegetation and dormant season to the annual fluxes (defined as 100%) are indicated above the bars

on the measurement days (Fig. 2a). It has been shown that stem $CO_2$ effluxes in boreal trees decrease significantly during winter periods, when stems are frozen[42]. Large amounts of $CO_2$ can nevertheless be released in short-term $CO_2$ burst events during freezing and thawing of tree stems and thus contribute significantly to the seasonal $CO_2$ dynamics[42]. We did not observe such bursts during autumn measuring campaigns when the air temperature was above zero, but slightly elevated stem $CO_2$ fluxes during February (pine) and March (birch, spruce) might indicate $CO_2$ bursts from freezing and thawing tree stems (Supplementary Fig. 1). The stem $N_2O$ flux dynamics, albeit at comparatively lower rates, follow a similar seasonality in spring (Fig. 1), thus supporting the idea that both $CO_2$ and $N_2O$ originate from a similar source. Both gases dissolved in the xylem sap might be released from the stem during freezing to avoid winter embolism in the xylem conduit and during thawing from the intercellular spaces, where gases can be trapped during the process of stem freezing[42].

On plots characterised by lower soil VWC, the stems even consumed $N_2O$ from the atmosphere during the dormant season, thus contributing to reduction of the annual source strength of trees and the ecosystem as a whole (Fig. 5a–c). Birch was identified as the strongest $N_2O$ sink. Dormant uptake by birch stems amounted to as much as 52% of the annual $N_2O$ emissions at the moderately wet plot (Fig. 5a). We speculate that the species variability in $N_2O$ exchange (Figs. 1, 5) might be explained by spatial variability of $N_2O$ concentration in soil, which is more pronounced under lower soil VWC. Under such conditions, $N_2O$ sources are more diverse due to simultaneously running aerobic and anaerobic N turnover processes leading to production and consumption of $N_2O$. Under dry conditions, therefore, root depth and distribution seem to play a more important role, species specificity is more pronounced, and differences among individual trees having different $N_2O$ sources available also are more prominent. This hypothesis should be confirmed by further research.

To the best of our knowledge, the limited number of studies reporting $N_2O$ exchange of tree stems present trees only as $N_2O$ emitters[13,14]. The only tree species known able consistently to take up $N_2O$ from the atmosphere is European beech[11]. That species' cryptogamic stem covers were shown to be organisms that might be co-responsible for beech's uptake of $N_2O$. In their study, Machacova et al. (2017)[11] observed that $N_2O$ consumption rates were closely related to the respiratory $CO_2$ fluxes of trees and cryptogams, thus indicating a connection between $N_2O$ consumption and the physiological activity of trees and microbial communities. Our observed $N_2O$ consumption by boreal tree stems is probably not linked to the physiological activity of the cryptogams associated with the tree bark, because during the dormant season any physiological activity in the forest is very low, as evidenced by the negligible stem $CO_2$ effluxes. We hypothesise that the reason for high $N_2O$ uptake observed in birch trees might be that birch trees, in contrast to the studied conifers, possess an aerenchyma system serving as a passive gas transport pathway within the tree[43]. Under low winter $N_2O$ concentration in soil, the broadleaf birch trees might hypothetically take up $N_2O$ from the atmosphere through lenticels in the bark, transport this gas along the concentration gradient into the roots, then perhaps release it into the soil at the root tips lacking exodermis. In contrast to the wood of conifers, that of the birches is a diffuse-porous type[44] that is more gas-permeable[45]. The $N_2O$ might also be reduced by denitrifiers directly in stem tissues below the lenticels, although such microbial activity would be decreased in wintertime[6,46]. At least in the case of *Betula potaninii*, it seems also that the vapour phase-based water and oxygen permeance of individual lenticels is significantly reduced

during wintertime due to the production by phellogen of compact tissues closing off lenticels at the end of the vegetation season. These tissues lacking intercellular spaces reduce gas exchange between the atmosphere and the system of intercellular spaces within the stem[47]. Hence, the mechanisms behind the uptake of $N_2O$ by trees and the fate of $N_2O$ remain unknown.

Similarly to the tree stem fluxes, the forest floor was a source of $N_2O$ in the vegetation season independently of the soil VWC (Fig. 5d). The rates of $N_2O$ emission were in line with those reported earlier for the same forest[48]. The similarly elevated emissions at all the studied plots in September (Fig. 1d) might be connected to litterfall, which is regarded as the largest external N input to the soil and hence suggested to stimulate $N_2O$ formation in the soil[48]. The forest floor fluxes subsequently decreased in the dormant season, which was in accordance with our earlier findings[48]. The effect of soil VWC on $N_2O$ fluxes was most pronounced during the dormant season, when $N_2O$ consumption was observed in the soils with low VWC ($0.21\,m^3\,m^{-3}$; i.e. dry plot; Fig. 5d). This $N_2O$ consumption reduced the annual forest floor $N_2O$ emissions by 40% at the dry plot (Fig. 5d). Nevertheless, forest floor $N_2O$ consumption was occasionally observed at all the plots throughout the year (Fig. 1d). Even though the $CO_2$ emissions from the forest floor also were significantly lower in the dormant season compared to the vegetation season (Supplementary Fig. 1d, 4d), these were still detectable.

In summary, during the dormant season, the tree stems and forest floor remain sources of $N_2O$ at plots characterised by high soil VWC, whereas tree stems act as $N_2O$ sinks under moderately wet soil water conditions. During the vegetation season, however, the soil VWC does not affect the $N_2O$ emissions from either trees or forest floor. Hence, our results highlight the need for winter flux measurements in order to correctly estimate the overall $N_2O$ budget of boreal forests.

**Boreal trees are net annual $N_2O$ emitters.** All the tree species studied were net sources of $N_2O$ at the annual scale (Fig. 6). To the best of our knowledge, this is the first study reporting annual course of $N_2O$ fluxes in boreal trees to include winter measurements. Neither the stem nor the forest floor $N_2O$ emissions were significantly influenced by the soil VWC at the annual scale (Fig. 1). Therefore, measurements at the wet and moderately wet plots, where all the species were present, were merged to evaluate the tree species-specific fluxes at the annual scale. The measurements at the dry plot were not included into this comparison because only pine trees were present there. Spruce was the strongest emitter of $N_2O$, with total emission per year of 0.91 mg $N_2O\,m^{-2}$ stem area and 2.4 g $N_2O\,ha^{-1}$ ground area, followed by pine (0.47 mg $m^{-2}$ and 1.7 g $ha^{-1}$) and birch (0.38 mg $m^{-2}$ and 0.71 g $ha^{-1}$) (Fig. 6). The forest floor emitted in total 9.4 mg $N_2O$ $m^{-2}$ soil area per year (i.e. 93.9 g $ha^{-1}$ per year), which is consistent with the annual $N_2O$ emissions of 8.8 mg $N_2O\,m^{-2}\,yr^{-1}$ estimated for the same forest during the years 2002–2003[48].

Based on the topographic wetness index (TWI) at the site, the dry plot represents 48%, the moderately wet plot 37%, and the wet plot 11% of the forest (remaining 4% accounting for standing water, Supplementary Fig. 5). Thus, we estimate that the annual emissions from the wet and moderately wet plots together represent ca 50% and the emissions from the dry plot ca 50% of the total forest fluxes, respectively. As we have demonstrated that the tree stem $N_2O$ fluxes are not controlled by soil water content at the annual scale, we confidently can conclude that the site type does not play a critical role in stem $N_2O$ fluxes.

Moreover, the differences in ecosystem level fluxes may result from tree species composition of the forest. As we found that spruce tree stems emitted significantly more $N_2O$ than did pine

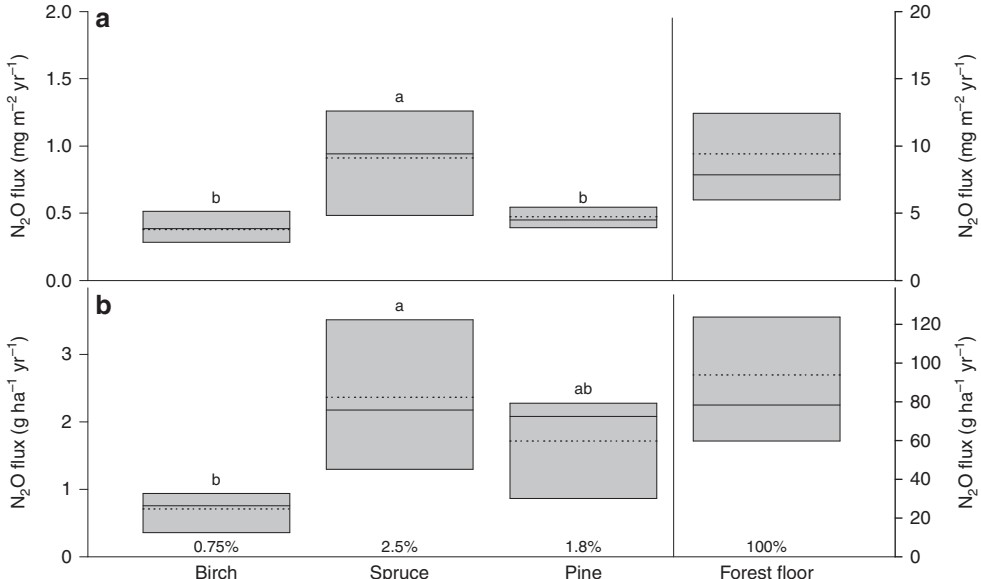

**Fig. 6** Annual $N_2O$ fluxes in tree stems and forest floor. The fluxes are expressed per stem or soil surface area unit (**a**) and scaled up to unit ground area of boreal forest (**b**). The fluxes are expressed as medians (solid line) and means (broken line) of measurements at both wet and moderately wet plots together, as the $N_2O$ fluxes did not vary significantly between those plots at the annual scale. The dry plot was not included into this comparison of annual fluxes because only pine trees were available at this plot. The stem fluxes were measured from six trees per species ($n = 6$), the forest floor fluxes were determined at six positions ($n = 6$). The box boundaries mark the 25th and 75th percentiles. Statistically significant differences in annual fluxes among birch, spruce and pine at $p < 0.05$ are indicated by different letters above the bars. The contributions of stem fluxes to forest floor $N_2O$ fluxes (equal to 100%) are expressed as percentages of the forest floor flux

**Table 1 Stand characteristics and tree biometric parameters**

|  | Tree height (m) | DBH (m) | Stem surface area (m²) | Forest density (trees ha⁻¹) | Stand basal area (m² ha⁻¹) |
|---|---|---|---|---|---|
| Birch (*Betula pendula* and *B. pubescens*) | | | | | |
| W plot | 12.3 ± 1.2 | 0.10 ± 0.01 | 1.9 ± 0.1 | 1200 | 6 |
| MW plot | 22.1 ± 0.7 | 0.21 ± 0.04 | 7.4 ± 1.2 | 200 | 4 |
| Spruce (*Picea abies*) | | | | | |
| W plot | 14.5 ± 4.4 | 0.17 ± 0.07 | 4.2 ± 2.3 | 400 | 4 |
| MW plot | 21.2 ± 1.0 | 0.24 ± 0.02 | 7.9 ± 1.0 | 400 | 5 |
| Pine (*Pinus sylvestris*) | | | | | |
| W plot | 18.2 ± 1.3 | 0.20 ± 0.03 | 5.8 ± 1.3 | 400 | 20 |
| MW plot | 20.6 ± 0.4 | 0.19 ± 0.01 | 6.1 ± 0.1 | 800 | 21 |
| D plot | 18.7 ± 0.7 | 0.19 ± 0.01 | 5.7 ± 0.6 | 1400 | 27 |

All variables (mean ± standard deviation) are related to birch, spruce, and pine trees at wet (W), moderately wet (MW), and dry (D) plots. The dry plot did not have spruce or birch trees. Trees were approximately 50 years of age, except for three birches and one spruce on the wet plot, which were unambiguously younger. The forest density is defined as number of individual trees per tree species and area unit of individual experimental plots. Stem diameter at breast height (DBH). Table modified after ref. [68]

and birch stems, spruce-dominated forests are predicted to emit more $N_2O$ than pine- or birch-dominated forests. The contribution of tree species to the forest floor $N_2O$ emissions was relatively low, however, amounting to 2.5, 1.8, and 0.75% for spruce, pine, and birch, respectively (Fig. 6) when the certain representation of tree species at each plot (Table 1) was included in upscaling. In Finland, Scots pine is the dominant tree species, accounting for 78% of forest land area coverage, while only 15% is covered by Norway spruce[49]. Our finding of the $N_2O$ emissions from spruce trees can be important for the estimation of $N_2O$ budget not only in boreal forests but also in temperate forests of Central Europe, where spruce is widely grown in monoculture[50]. Spruce trees' stronger capability to exchange $N_2O$ with the atmosphere may be related to their physiological activity. In our study, spruces had the highest projected leaf area per tree (88 m² on average) among the tree species studied (birch 54 m², pine 28 m²). Greater leaf area results in larger amounts of $CO_2$

assimilated and $H_2O$ transpired per spruce tree than per pine or birch tree. The greater physiological activity of spruce is further reflected in the higher annual sum of stem $CO_2$ efflux amounting to 0.867 kg $CO_2$ m⁻² and 2303 kg ha⁻¹, compared to 0.590 kg $CO_2$ m⁻² (2140 kg ha⁻¹) and 0.427 kg $CO_2$ m⁻² (738 kg ha⁻¹) for pine and birch trees, respectively (Supplementary Fig. 6). Ge et al.[51] presented the same conclusion from their study of different boreal tree species. The variation in $N_2O$ emission rates among plant species also can result from plants' effects on soil $N_2O$ production and consumption, which can themselves differ significantly among species, rather than from different transpiration rates or direct plant production of $N_2O$ in plant tissues[52]. Although deciduous tree species tend to increase soil $N_2O$ production more so than do conifers[29,30], the effects of individual tree species are not uniformly presented among studies. Further research is therefore needed to understand the observed differences in $N_2O$ emission rates among tree species.

Lack of canopy level $N_2O$ flux measurements brings additional uncertainty in the forest ecosystem $N_2O$ budget. Based on our previous research, we have shown that the leaf emissions by pine trees could exceed those of stems by as much as 16 times[13]. We therefore expect that boreal tree species might contribute even more significantly to the forest $N_2O$ exchange. Although measurements of above-canopy $N_2O$ exchange in forest ecosystems using such micrometeorological techniques as eddy covariance, eddy accumulation, or flux gradient methods have been used only rarely[53–55], these could improve our view in the future.

We have demonstrated that $N_2O$ emissions from tree stems are driven by physiological activity of the trees and by ecosystem activity, showing higher emissions during the active growing period and variation between uptake and emissions during the dormant season. Although our study may well be applicable to large upland forest areas in the boreal zone, which are typically N limited[56], our findings may not apply directly in N-affected central European or American forests known to exhibit elevated soil $N_2O$ emissions due to higher soil N content and faster N turnover rates[57–59]. The N status of a forest directly influences soil $N_2O$ concentration, which has been shown to be a good proxy for $N_2O$ transport via the transpiration stream of trees[8]. Until more studies and process understanding emerge, the global strength of $N_2O$ emissions from trees will remain largely unknown and could possibly be estimated by, for example, adding a fixed percentage (e.g. 10%) to the forest floor $N_2O$ emissions to represent $N_2O$ emission from trees.

In summary, we have shown that all widespread boreal tree species are net annual sources of $N_2O$, with spruce being the strongest emitter. The highest stem emissions were detected during summer, but remained detectable also during winter dormancy. Seasonal changes in $N_2O$ fluxes tightly correlate with changes in $CO_2$ fluxes, which are particularly driven by temperature and light intensity. The physiological processes of trees thus could be used as indicators of the $N_2O$ flux dynamics in boreal forests. Furthermore, our results unequivocally indicate necessity to include seasonality of $N_2O$ exchange into global models and forest ecosystem process models to determine comprehensive total flux estimates.

## Material and methods

**Site description and experimental design.** The measurements were performed in boreal forest near the SMEAR II station (Station for Measuring Ecosystem–Atmosphere Relations) at Hyytiälä, southern Finland (61°51′ N, 24°17′ E, 181 m a.s.l.) from June 2014 until May 2015. The long-term annual mean temperature and precipitation for this site are 3.5 °C and 711 mm, respectively[60]. Seasonal courses of basic environmental variables during the period studied are shown in Fig. 2. The highest mean daily air temperatures within the forest stand were 24 °C in July and 23 °C in August and the lowest −18 °C in December and −15.5 °C in January (Fig. 2a). Soil temperature in the C1-horizon corresponding to the depth of 38–60 cm was lowest in February and March (between 1.7 and 1.9 °C) and highest in August (12.2 °C, Fig. 2b). There was a continuous snow cover between mid-December and early April (Fig. 2b). Soil VWC (Fig. 2c) in the C1-horizon was lowest in August, remained rather constant between November and April, but was highest in March. The seasonal course for the daily sum of photosynthetically active radiation (PAR; Fig. 2d) has a pattern typical for boreal ecosystems.

The studied mixed forest is dominated by *P. sylvestris*, with *P. abies*, *B. pubescens*, and *B. pendula* as other abundant species. The forest is heterogeneous in soil type and characteristics (Haplic podzol on glacial till with some paludified organic soil), tree

species and understorey vegetation composition, and forest structure. Such heterogeneities accordingly lead to heterogeneity in soil VWC (Supplementary Fig. 5).

Three plots naturally differing in soil VWC—wet (0.81 ± 0.02 $m^3 m^{-3}$; mean ± standard error), moderately wet (0.40 ± 0.02 $m^3 m^{-3}$), and dry (0.21 ± 0.01 $m^3 m^{-3}$)—were selected for this study. Three representative trees of pine, spruce, and birch were chosen per plot ($n = 3$) for stem flux measurements, except that at the dry plot only *P. sylvestris* was present. Characteristics of the representative trees are shown in Table 1. Forest floor $N_2O$ and $CO_2$ fluxes were measured at three representative positions on both the wet and moderately wet plots ($n = 3$) and at six positions on the dry plot ($n = 6$). The ground vegetation at the wet plot was dominated by *Sphagnum* sp. followed by *Polytrichum commune* and *Equisetum sylvaticum*. Some *Comarum palustre*, *Trientalis europae*, and *Carex digitata* were also present in the soil chambers. The dominating species in the chambers at the moderately wet plot were *Hylocomium splendens*, *P. commune*, and *Sphagnum sp*. Also common were *Pleurozium schreberi*, *Dicranum polysetum*, *Vaccinium vitis-idaea*, *Vaccinium myrtillus*, and *Rubus saxatilis*. At the dry plot, the dominant species were *V. vitis-idaea*, *V. myrtillus*, *H. splendens*, and *P. schreberi*.

Stem and forest floor fluxes of $N_2O$ and $CO_2$ were measured simultaneously at each plot. Acquiring one measurement set from all three plots required 1–2 weeks, and all the plots were measured at least once per month. The most intensive measurement periods were the main vegetation season (June–August), when the flux measurements were repeated six times, and February, with two repetitions. Adverse conditions did not allow flux measurements in November and January, and thus fluxes for these months were estimated using a linear interpolation of fluxes from adjacent months. All fluxes were determined between 10 a.m. and 4 p.m. of non-precipitating days to prevent possible effect of diurnal cycle. The means of air and soil temperature, soil VWC, and PAR for flux measurement days of each month are shown in Fig. 2.

**Stem flux measurements.** The fluxes of $N_2O$ and $CO_2$ at the bottom part of the stems (approximately 20 cm above the soil) were measured manually using two different types of static chamber systems: box and circumferential chambers. The chambers were installed in May 2014. The box chambers[11,61] consisted of transparent plastic containers with removable air-tight lids (Lock & Lock, Anaheim, CA, USA) and a neoprene sealing frame. They were gas-tightly affixed to the carefully smoothed bark surface using silicone. Per each tree, two rectangular box chambers (total area of 0.0176 $m^2$ and total internal volume of 0.0012 $m^3$) were installed at one stem height on opposite sides of the stem and interconnected with polyurethane tubes into one flow-through chamber system (for more details see Machacova et al.[11]). The circumferential chambers were as described by Machacova et al.[13] with slight modification. The chambers consisted of a wire skeleton and a tube-fitting brace for inlet and outlet connectors which were wrapped six times with a plastic stretch foil to form the chamber wall. The top and bottom ends of the foil were sealed with neoprene, silicone, and cable ties to the carefully smoothed bark (for details see Machacova et al.[13]). The internal volume of the circumferential chambers ranged between 0.0030 and 0.0054 $m^3$ and the stem surface area covered by the chamber ranged between 0.083 and 0.258 $m^2$, depending on stem diameter. Mixing of the air inside both chamber systems was produced via air circulation by gas pumps (DP0140/12 V, Nitto Kohki, Tokyo, Japan; NMP 850.1.2., KNDC B, KNF Neuberger, Freiburg, Germany). Fans (412 FH, ebm-papst, Mulfingen, Germany; KF0410B1H, Jamicon, Kaimei

Electronic Corp., New Taipei City, Taiwan) in circumferential chambers enhanced the mixing process. The two different chamber types were equally distributed among the studied tree species, although box chambers were mostly installed on spruces because an abundance of lower branches typical for spruces prevented installation of circumferential chambers. The results obtained from the two different stem chamber systems were comparable. Gas-tightness of all the chambers was regularly tested throughout the year.

For the flux measurements, nine gas samples (each 20 mL) were taken via a septum at 0, 30, 60, 90, 130, 170, 220, 270, and 320 min after closure of the chamber system and stored in pre-evacuated gas-tight glass vials (Labco Exetainer, Labco, Ceredigion, UK). The possible changes in air pressure within chambers were compensated by the flexible wall reducing an internal volume of circumferential chambers and by simultaneous injection of ambient air into the box system. The box chambers were left open between the measuring campaigns, but the circumferential chambers were left closed between the measurements on following days. In the latter case, the chamber headspace was washed with ambient air for at least 30 min prior to the flux measurement. The monitored $CO_2$ concentration in the chambers provided an indicator as to when the washing of the chambers was sufficient.

**Forest floor flux measurements**. Forest floor $N_2O$ and $CO_2$ fluxes were measured using two soil chamber types differing in their internal volumes. At the wet and moderately wet plots, three large manual aluminium opaque chambers (internal volume between 0.092 and 0.140 $m^3$, depending on vegetation cover, soil surface area of 0.298 $m^2$) described as chamber n. 13 by Pihlatie et al.[62] were used. At the dry plot, the forest floor fluxes were determined by six smaller manual stainless steel opaque chambers (internal volume 0.020 $m^3$, soil surface area 0.116 $m^2$)[48]. The collars of all chambers were installed in 2013 or earlier in the vicinity of the investigated trees. A fan (412 FH, ebm-papst, Germany; KF0410B1H, Kaimei Electronic Corp., Taiwan) was used to mix the air inside all chambers.

The large soil chambers were closed for ca 75 min during which gas samples (65 mL each) were taken at time intervals of 1, 5, 15, 25, 35, 55, and 75 min. The gas sampling intervals from the small soil chambers at the dry plot were set to 1, 5, 15, 30, and 45 min after the time of closure. All gas samples were transferred from syringes to glass vials (Labco Exetainer) until the analysis by gas chromatograph.

**Gas analyses**. The gas samples were stored in gas-tight glass vials at 7 °C and analysed by an Agilent 7890 A gas chromatograph (Agilent Technologies, Santa Clara, California, USA)[62]. The gas chromatograph was equipped with an electron capture detector for $N_2O$ analyses, and a flame ionisation detector for $CO_2$ detection. The electron capture detector (operating at 380 °C) was supplied with argon/methane (15 mL min$^{-1}$) as a make-up gas. The flame ionisation detector (operating at 300 °C) was run with synthetic air (450 mL min$^{-1}$) and hydrogen (45 mL min$^{-1}$), with nitrogen (5 mL min$^{-1}$) as a make-up gas. Helium (45 mL min$^{-1}$) was used as a carrier gas. Porapak Q 80-100 and HayeSep Q 80-100 mesh columns (Agilent Technologies) were used for water vapour removal and gas separation. Oven temperature was maintained at 60 °C during analyses. The gas samples were injected automatically using a GX-271 autosampler (Gilson Liquid Handler, Middleton, Wisconsin, USA). ChemStation B.03.02 software (Agilent Technologies) was used to control the chromatograph system and analyse the data.

The $N_2O$ and $CO_2$ concentrations in gas samples were calculated based on a four-point concentration calibration curve (0.31, 0.34, 0.36, and 0.39 μmol $N_2O$ mol$^{-1}$; 400, 733, 1067, and 1500 μmol $CO_2$ mol$^{-1}$). The standards were analysed at the beginning of each run and after every ca 27 gas samples. Standard samples (0.34 μmol $N_2O$ mol$^{-1}$, 733 μmol $CO_2$ mol$^{-1}$) were applied after every ca 9–18 samples to detect possible drifts occurring during the analysis.

**Flux calculation**. The stem and forest floor fluxes were quantified based on the linear $N_2O$ and $CO_2$ concentration changes in the chamber headspace over time (for the equation, see Machacova et al.[13]). The stem and forest floor monthly fluxes were calculated as mean daily fluxes of a given month multiplied by the number of days for each of the months. The annual and seasonal fluxes were calculated as sums of monthly fluxes. Missing fluxes were estimated using linear interpolation of fluxes from adjacent months (for November, January) or as the mean of flux rates simultaneously detected from adjacent chambers. The annual fluxes were further scaled up to those for 1 ha of a boreal forest with characteristics of the studied plots. An extrapolation was based on mean stem surface area, tree density, and stand basal area, all estimated for each individual tree species and plot (Table 1). The upscaling procedure is described in Machacova et al.[13].

**Ancillary measurements**. Interpretation of the flux data was based on the seasonal dynamics of environmental parameters, including in particular soil water content and temperature, air temperature, and radiation intensity. The soil water content and soil temperature were measured close to the studied trees and soil chambers in the A-horizon (0–5 cm soil depth) using a manual HH2 Moisture Meter with Theta Probe (ML2x, AT Delta-T Devices, Cambridge, UK) and continuously running DS1921G Maxim Thermochron iButtons (Maxim Integrated, San Jose, California, USA), respectively. In addition, the following environmental and ecosystem parameters, which continuously are determined at the SMEAR II station, were used for further non-parametric correlation and PLS path analyses: soil water content (TDR100, Campbell Scientific, North Logan, Utah, USA) and soil temperature (Philips KTY81/110, NXP, Eindhoven, Netherlands) in the C1-horizon (38-60 cm soil depth)[63,64]; air temperature at 8 m height within the forest stand (Pt100 sensors, with radiation shield by Metallityöpaja Toivo Pohja); relative humidity at 16 m height (MP102H RH sensor, Rotronic AG, Bassersdorf, Switzerland); PAR at 18 m height (Li-190SZ, Li-Cor, Lincoln, Nebraska, USA); UV A and B radiation at 18 m height (SL501A radiometer, Solar Light Company, Glenside, Pennsylvania, USA); net ecosystem exchange of $CO_2$ (measured by eddy covariance method[65]); GPP (derived from net ecosystem exchange); and evapotranspiration at 23 m height (measured by eddy covariance method, with gaps in the measured $H_2O$ flux time series filled using linear regressions of flux on net radiation[66,67]).

**Statistics**. The flux data were tested for normal distribution (Shapiro–Wilk test) and equality of variances in different sub-populations; $t$-test and one-way analysis of variance for multiple comparisons were applied for normally distributed data. Non-parametric Mann–Whitney rank sum test and Kruskal–Wallis one-way analysis of variance on ranks for multiple comparisons were applied for non-normally distributed data and/or data with unequal variances. The $n$ values for statistical analyses are stated in the figure legends. Detailed non-parametrical Spearman correlation analyses and PLS path modelling of $N_2O$ fluxes were carried out. Hence, tree stem and soil $N_2O$ fluxes were compared

to $CO_2$ fluxes, soil temperature and soil water content measured adjacent to the trees and soil chambers, and to the aforementioned environmental and ecosystem parameters continuously measured at the SMEAR II station. Statistical significance of all tests was defined at $p < 0.05$. The statistics were run using SigmaPlot 11.0 (Systat Software, San Jose, California, USA), Python scripts and SmartPLS 3 software (SmartPLS, Boenningstedt, Germany).

### Data availability

The datasets generated and analysed during this study are available from the authors upon reasonable request.

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

## Acknowledgements
This research was supported by the Czech Science Foundation (17-18112Y), the Ministry of Education, Youth and Sports of the Czech Republic within the National Sustainability ProgramI (grant number LO1415), EU FP7 project ExpeER (Grant Agreement 262060), Emil Aaltonen Foundation, Academy of Finland Research Fellow projects (292699, 263858, 288494), The Academy of Finland Centre of Excellence (projects 1118615, 272041), ICOS-Finland (281255), and the European Research Council (ERC) under the European Union's Horizon 2020 research and innovation programme (Grant Agreement 757695). We thank Marian Pavelka, Jiří Dušek, Stanislav Stellner, Jiří Mikula, Marek Jakubík, Janne Levula and Matti Loponen for technical support and Uwe Großmann for IT support in data processing.

## Author contributions
K.M. and M.P. had the idea for the study. K.M., M.P. and E.V. designed the study. K.M. and E.V. carried out the field measurements and analysed the data. K.M., M.P., E.V. and O.U. contributed to writing the manuscript.

## Competing interests
The authors declare no competing interests.
