## [Peer Review File · Nature Communications]

Reviewers' comments:

Reviewer #1 (Remarks to the Author):

I have enjoyed reading the manuscript. It describes contributions of three boreal tree species to N₂O exchange.

It is an important and interesting piece of the puzzle of the greenhouse gasses dynamics. The production of the N₂O by trees have been shown in previous studies (cited in the manuscript) but the paper presents a comprehensive assessment of yearly budget as well as seasonal pattern of N₂O dynamics. It shows the need to include the N₂O tree dynamics in the ecosystem calculations. At the same time it highlights the variability among the different species. The N₂O stem emission were closely related to CO₂ stem emissions and to a lesser degree to gross primary productivity. The N₂O emissions also correlated with air temperature and PAR intensities. The study highlights the important of considering seasonality especially in the boreal biome where it is so pronounced. Evidence of the trees ability to take up N₂O, although only under certain conditions, is an interesting point.

The study is rigorous, well designed and analysed.

Figures are clear and informative. The outcomes are convincing and well supported.

Still the clarity of the message could be improved, by choosing and reinforcing the main message.

The trees acted consistently as N₂O source on wet sites, but the exchange was more variable on drier sites. Can you comment further on the variability of among different site types and on the distribution of the sites in the boreal forest? Can you extrapolate from surveys/map extent of dry and wet sites and the implications for prevailing dynamics?

What is the applicability of the results in the rest of the range of the boreal forest?

The authors state that although the N₂O production by the boreal trees is quite small (compared to temperate trees) the implications for global atmospheric balance may be large due to the large area of the boreal forests. It is however not clear how representative is the observed dynamics for the rest of the boreal region. How does the climate in the majority of the Siberian and Canadian boreal forest compare to the climate in the Finish sites? The dynamic was shown to be influenced by the climatic variables. The temperatures are likely to be more consistently below zero in large areas of the region. Are the N₂O exchanges from trees considered in the global models now at all? What is best guidance for the accounting for this source at the moment? Could it be approximated form CO₂ measurements? Or e.g. Adding 10% to forest floor emission estimates? Can you discuss relatively small contribution of the trees to the effort of obtaining more accurate estimates?

These implications should be considered and explained so that the study is of wide interest and globally applicable.

Minor editing is needed to improve grammar and flow of the text .

Reviewer #2 (Remarks to the Author):

General comments

In general, the manuscript is very well written and easy to follow. The experiments appear to have been well carried out and the results are convincing. Although I found it quite large at first glance the extend of the manuscript is adequate and covers –at least to my knowledge- all aspects of stem-atmosphere gas exchange. It has improved my knowledge about gas exchange of trees (excluding emissions from leaves, so the picture is rather incomplete, but this is made clear by the authors) and their role in the ecosystem greenhouse gas budget. The dataset is sufficient to draw the conclusions of the manuscript. The given correlations between N₂O gas exchange and physiological and environmental parameters are not very surprising but reasonable. My only concern is the number of figures. Ten figures in this manuscript is quite a lot, I recommend to shift Fig. 4 to 6 to a supplementary section, as the outcome of the statistical analysis is also given in the text.

The manuscript is of high quality and deserves publication in a high-ranking journal like NCMSS.

Therefore, I recommend publication of this manuscript with minor revisions.

Minor comments

At many places (Lines 39, 44, 221, 250, 408, 416] the cited reference is given as [ref. 2] for example. Please change to superscript numbers.

Line 117-119: Please rephrase this sentence.

Line 126-127: You said that the stem and forest flow could be decoupled. Could it be possible that mycorrhizal fungi are involved either in dissolved N₂O uptake or N₂O production and thereby trigger stem N₂O exchange?

Line 180: You specify CO₂ burst events. Similar burst events are known for N₂O due to freezing-thawing cycles. I would have expected them to in the forest floor and tree stem emissions. To you have an explanation for the absence of such N₂O burst emissions in your experiment?

Reviewer #3 (Remarks to the Author):

Commenting "Seasonal variation in stem N₂O exchange of boreal trees relates to their physiological activity" submitted by Dr. Katerina Machacova et al.

Boreal forests are widespread, where nitrous oxide has been largely overlooked. The authors spent a lot of time to measure annual changes in soil and stem nitrous oxide fluxes of the common boreal tree species spruce, birch, and pine. Particularly, the greenhouse gas fluxes were measured in the winter. I approve the authors worked hard and pushed this topic forward and stored useful data. However, I have the following concerns.

Stem nitrous oxide fluxes have been previously widely investigated. This work is a case investigation and lacks a novelty. When I am a reader, I cannot be impressed by the results.

What is physiological activity? What are physiological parameters? Soil water content, soil and air temperatures, and photosynthetically active radiation? To my knowledge, the authors measured environmental factors but did not measure tree physiological parameters. I do not think the authors studied the physiological activity and its direct relationships with seasonal nitrous oxide fluxes.

The purpose of this study is seasonal changes in stem nitrous oxide fluxes. Why are almost half of figures on carbon dioxide? I suggest that the figures on carbon dioxide can be deleted.

This work did not study the mechanisms of nitrous oxide production and emissions. The authors provided too much speculative discussion without evidence. I suggest that the discussion can be largely shortened and improved.

Point-by-point response to the referees' comments

Revised version

MS ID: NCOMMS-18-29103

MS Title: 'Seasonal variation in stem N₂O exchange of boreal trees relates to their physiological activity'

MS Authors: Katerina Machacova, Elisa Halmeenmäki, Otmar Urban & Mari Pihlatie

Reviewer #1 (Remarks to the Author):

I have enjoyed reading the manuscript. It describes contributions of three boreal tree species to N₂O exchange.

*It is an important and interesting piece of the puzzle of the greenhouse gasses dynamics. The production of the N₂O by trees have been shown in previous studies (cited in the manuscript) but the paper presents a **comprehensive assessment of yearly budget as well as seasonal pattern of N₂O dynamics**. It shows the need to include the N₂O tree dynamics in the ecosystem calculations. At the same time it highlights the **variability among the different species**. The N₂O stem emission were **closely related to CO₂ stem emissions and to a lesser degree to gross primary productivity**. The N₂O emissions also correlated with air temperature and PAR intensities. The study highlights the important of considering seasonality especially in the boreal biome where it is so pronounced. **Evidence of the trees ability to take up N₂O**, although only under certain conditions, is an interesting point.*

The study is rigorous, well designed and analysed.

Figures are clear and informative. The outcomes are convincing and well supported.

*Still the clarity of the message could be improved, by **choosing and reinforcing the main message**.*

- We want to thank the reviewer for the suggestions to focus in on and reinforce the main message of the manuscript. We have now carefully read through and analysed what we consider to be the manuscript's main message. In the revised version of the manuscript we have carefully highlighted the following points. For the first time, we present 1) a comprehensive assessment of yearly budget and seasonal and monthly pattern of N₂O flux dynamics in main representatives of boreal tree species and soil. Also, we report 2) high variability in tree stem N₂O dynamics among the different tree species, and that 3) N₂O stem emission were closely related to the physiological activity of trees and ecosystem – mainly to stem CO₂ efflux and gross primary production (GPP). We also want to highlight 4) the evidence of the trees' ability to take up N₂O. Finally, we estimate 5) the contribution of tree stem N₂O flux to the net ecosystem N₂O exchange. We improved the manuscript accordingly. These main results are now carried through the entire manuscript as well as carefully discussed. We believe that modification of the manuscript based on the comments of all reviewers improved the clarity of the main messages of the manuscript (please, see the proposed modifications throughout the manuscript in response to all reviewers' comments).
- Discussion on the observed high variability in tree stem N₂O dynamics among the different tree species has been re-worded and improved in many places of the revised version of the manuscript (e.g. page 5, rows 192–216, page 6, rows 257–282).
- We elaborated in the Results (page 6, rows 237–249) that the stem N₂O fluxes of pine, spruce, and birch trees were not significantly different at annual scale between trees growing on plots characterized with different soil water content, and, as soil moisture was not found to be among the driving variables of tree stem N₂O fluxes, we can confidently conclude that soil moisture is not

critically affecting stem N₂O fluxes. Hence the site type difference in our study area does not play an important role, as stated in page 6, rows 250-256.

- We added information about tree species composition of boreal forests in Finland, and discussed the role of spruce trees as the greatest emitter of N₂O in Finland and in Europe generally (page 6, rows 257–282).
- Furthermore, in order to relate our results to a more general/global context, we added brief discussion on the topic of how representative are the studied N₂O flux dynamics within boreal and temperate regions (N limited vs. N “saturated” forest ecosystems) (page 7, rows 290–301).
- Finally, we edited our overall conclusions in the manuscript to improve the clarity of the main message/results of our study (i.e. answering of the research questions stated at the end of the introduction section; see page 7, rows 302–310).

The trees acted consistently as N₂O source on wet sites, but the exchange was more variable on drier sites. Can you comment further on the variability of among different site types and on the distribution of the sites in the boreal forest? Can you extrapolate from surveys/map extent of dry and wet sites and the implications for prevailing dynamics?

- We added a new Supplementary Fig. 5 showing the topographic wetness index (TWI) of the studied forest site. We newly estimated the contribution of tree stem N₂O fluxes from the studied plots characterized with different soil moisture to the total forest fluxes (page 6, rows 250–256): “Based on the topographic wetness index (TWI) at the site, the dry plot represents 48%, the moderately wet plot 37%, and the wet plot 11% of the forest (remaining 4% accounting for standing water, Supplementary Fig. 5). Thus, we estimate that the annual emissions from the wet and moderately wet plots together represent ca 50% and the emissions from the dry plot ca 50% of the total forest fluxes, respectively. As we have demonstrated that the tree stem N₂O fluxes are not controlled by soil water content at the annual scale, we confidently can conclude that the site type does not play a critical role in stem N₂O fluxes.”
- The issue of the inter-site variability in tree stem N₂O fluxes is newly, briefly discussed on page 4–5, rows 184–191: “We speculate that the species variability in N₂O exchange (Figs. 1, 5) might be explained by spatial variability of N₂O concentration in soil, which is more pronounced under lower soil VWC. Under such conditions, N₂O sources are more diverse due to simultaneously running aerobic and anaerobic N turnover processes leading to production and consumption of N₂O. At dry conditions, therefore, root depth and distribution seem to play a more important role, species specificity is more pronounced, and differences among individual trees having different N₂O sources available also are more prominent. This hypothesis should be confirmed by further research.”

What is the applicability of the results in the rest of the range of the boreal forest?

The authors state that although the N₂O production by the boreal trees is quite small (compared to temperate trees) the implications for global atmospheric balance may be large due to the large area of the boreal forests. It is however not clear how representative is the observed dynamics for the rest of the boreal region. How does the climate in the majority of the Siberian and Canadian boreal forest compare to the climate in the Finish sites? The dynamic was shown to be influenced by the climatic variables. The temperatures are likely to be more consistently below zero in large areas of the region.

- We want to thank the reviewer for this important comment. We have now added discussion on the representativeness of our measurement site within Finland and generally in the boreal region (see pages 6–7). Based on an analysis of the climate statistics between Finland (our site) and the Canadian boreal forest region, we find very similar seasonality (min and max temperatures) and mean annual temperature. Although the climate statistics of Finnish and Canadian boreal forests as well as seasonality in temperatures are very similar, upscaling from one representative study site to

the whole boreal forest region seems too uncertain. We write this in the paper, but we also want to state that based on our data it is extremely difficult to extrapolate to other boreal forest regions (page 7, rows 290–301) as follows: “We have demonstrated that N₂O emissions from tree stems are driven by physiological activity of the trees and by ecosystem activity, showing higher emissions during the active growing period and variation between uptake and emissions during the dormant season. Although our study may well be applicable to large upland forest areas in the boreal zone, which are typically nitrogen (N) limited (Högberg et al., 2017), our findings may not apply directly in N-affected central European or American forests known to exhibit elevated soil N₂O emissions due to higher soil N content and faster N turnover rates (Aber et al., 1998; Butterbach-Bahl et al. 2002; Kreutzer et al., 2010). The N status of a forest directly influences soil N₂O concentration, which has been shown to be a good proxy for N₂O transport via the transpiration stream of trees (Pihlatie et al., 2005). Until more studies and process understanding emerge, the global strength of N₂O emissions from trees will remain largely unknown and could possibly be estimated by, for example, adding a fixed percentage (e.g. 10%) to the forest floor N₂O emissions to represent N₂O emission from trees.” (see also response to the next reviewer’s comment)

- Although we considered that adding a comparison of weather statistics between the measurement site and Canadian and Siberian boreal forests is not necessary, we are presenting them here in support of our decision: Long-term weather statistics at our measurement site (Hyytiälä, Finland), with mean annual temperature of 3.5 °C and annual precipitation of 711 mm (1981–2020), and those in the boreal forest region in Canada (http://climate.weather.gc.ca/climate_normals/index_e.html), show that the Hyytiälä site very well represents the boreal climate of the Canadian forests. As a result, we can state that the seasonal N₂O dynamics at the Hyytiälä SMEAR II site are representative of other boreal forests in the world. Inasmuch as our study is the first to focus on the seasonal variation, however, more research is clearly needed to determine whether the dynamics are similar in other boreal forests of Scandinavia, Siberia and Canada.

*Are the N₂O exchanges from trees considered in the global models now at all? What is best guidance for the accounting for this source at the moment? Could it be approximated from CO₂ measurements? Or e.g. Adding 10% to forest floor emission estimates? Can you discuss relatively small contribution of the trees to the effort of obtaining more accurate estimates?
These implications should be considered and explained so that the study is of wide interest and globally applicable.*

- It is our understanding that the N₂O exchange of trees is currently not considered in either global models or any forest ecosystem process models. This is because the research topic is very new and the number of published papers dealing with trees and N₂O dynamics is rather small. Also, this is the first study to demonstrate a strong seasonality, which would be relevant to any process models and also should be acknowledged in the global models. We feel that this research field is still too new to include these data into global N₂O models, partly because we lack a process for understanding the N₂O exchange of trees. Our study plays a critical role in adding to understanding of the drivers (i.e. physiological activity of the trees and the ecosystem) of N₂O fluxes of tree stems. Including the dependencies of tree stem N₂O fluxes and stem CO₂ efflux, ecosystem evapotranspiration, and GPP into N cycling process models, or more likely into statistical models, could serve as the first steps towards estimating the rate of N₂O emissions from trees. Nevertheless, estimation of regional or global N₂O emissions from trees would require understanding of regional drivers of tree-N₂O dynamics. An intensive European research project on N oxide emissions from European forest ecosystems (NOFRETETE) has concluded that the regional estimates of N₂O emissions from European forest soils depend on the nutrient status, N deposition, and soil type, while at local scale the soil N₂O emission dynamics were driven by soil moisture and temperature (Pilegaard et al. 2006). More field data for the driving variables are urgently needed if we are to obtain regional or global estimates of tree N₂O emissions. If N₂O exchange is measured from different latitudes, vegetation zones, forests and tree species of varying N status, these together could identify

relationships between the driving variables (climate, N status/N deposition, mean annual temperature, dormancy period, soil N₂O fluxes, etc.).

- We have added a discussion concerning the current understanding of tree N₂O dynamics (pages 6–7) and uncertainties in global N₂O estimates to page 7, rows 298–301: “Until more studies and process understanding emerge, the global strength of N₂O emissions from trees will remain largely unknown and could possibly be estimated by, for example, adding a fixed percentage (e.g. 10%) to the forest floor N₂O emissions to represent N₂O emission from trees.”

References

- Aber, J. et al. Nitrogen saturation in temperate forest ecosystems – Hypotheses revisited. *Bioscience* **48**, 921–934, 1998.
- Butterbach-Bahl, K., Gasche, R., Willibald, G. & Papen, H. Exchange of N-gases at the Högwald Forest – A summary. *Plant Soil* **240**, 117–123, 2002.
- Högberg, P., Näsholm, T., Franklin, O. & Högberg, M.N. Tamm Review: On the nature of the nitrogen limitation to plant growth in Fennoscandian boreal forests. *Forest Ecol. Manag.* **403**, 161–185, 2017.
- Kreutzer, K., Butterbach-Bahl, K., Rennenberg, H. & Papen, H. The complete nitrogen cycle of an N-saturated spruce forest ecosystem. *Plant Biology* **11**, 643–649, 2009.
- Pihlatie, M., Ambus, P., Rinne, J., Pilegaard, K. & Vesala, T. Plant-mediated nitrous oxide emissions from beech (*Fagus sylvatica*) leaves. *New Phytol.* **168**, 93–98, 2005.
- Pilegaard, K., et al. Factors controlling regional differences in forest soil emission of nitrogen oxides (NO and N₂O). *Biogeosciences* **3**, 651–661, 2006.

Minor editing is needed to improve grammar and flow of the text.

- The entire manuscript was carefully edited by Gale A. Kirking, Editor-in-Chief at the professional editing firm English Editorial Services (www.englisheditorialservices.com). Mr. Kirking is a native speaker of English with many years of editorial experience in the life sciences area.

Reviewer #2 (Remarks to the Author):

General comments

In general, the manuscript is very well written and easy to follow. The experiments appear to have been well carried out and the results are convincing. Although I found it quite large at first glance the extend of the manuscript is adequate and covers –at least to my knowledge- all aspects of stem-atmosphere gas exchange. It has improved my knowledge about gas exchange of trees (excluding emissions from leaves, so the picture is rather incomplete, but this is made clear by the authors) and their role in the ecosystem greenhouse gas budget. The dataset is sufficient to draw the conclusions of the manuscript. The given correlations between N₂O gas exchange and physiological and environmental parameters are not very surprising but reasonable.

My only concern is the number of figures. Ten figures in this manuscript is quite a lot, I recommend to shift Fig. 4 to 6 to a supplementary section, as the outcome of the statistical analysis is also given in the text.

- We want to thank the reviewer for constructive comments that certainly have improved the manuscript.
- We carefully considered each of the figures in the manuscript and have now moved Fig. 2 (Annual course of monthly and annual CO₂ fluxes from stems of birch, spruce, and pine, and from forest floor...), Fig. 4 (Partial least squares path modelling of N₂O fluxes in stems of birch for 2014–2015),

Fig. 6 (Partial least squares path modelling of N₂O fluxes in stems of pine for 2014–2015), Fig. 8 (CO₂ fluxes in stems of birch, spruce and pine, and in forest floor at annual scale, for vegetation season, and for dormant season), and Fig. 10 (Annual CO₂ fluxes in stems of birch, spruce, and pine, and in forest floor...) to the supplementary section.

- With respect to the Figs 4–6 (partial least squares path modelling of N₂O fluxes in stems of different tree species), we moved the Figs 4 and 6 (birch and pine) to the supplementary material, and left Fig. 5 (spruce) in the paper to show the possible mechanisms of stem N₂O fluxes, as also requested by Reviewer 3. These pathway analyses show rather similar results for all three tree species, which fact supports our decision to show just one of them in the main paper. While Figs. 2, 8 and 10 show CO₂-related results that support our main findings in the manuscript, the main message is described also in the manuscript and so the figures can be moved to the Supplementary material (also in accordance with the comments of Reviewer 3).

The manuscript is of high quality and deserves publication in a high-ranking journal like NCMSS. Therefore, I recommend publication of this manuscript with minor revisions.

Minor comments

At many places (Lines 39, 44, 221, 250, 408, 416) the cited reference is given as [ref. 2] for example. Please change to superscript numbers.

- We are aware of this issue. To our knowledge, [ref. X] format is often used when following text string ending with a number/exponent (such as y⁻¹, N₂ ...) in order to avoid misinterpretation of the units or chemical compounds in such examples. We changed the cited references to superscript numbers in most cases, e.g. when attached to a year (e.g. page 8, rows 365-371).

Line 117-119: Please rephrase this sentence.

- This sentence was corrected

Line 126-127: You said that the stem and forest flow could be decoupled. Could it be possible that mycorrhizal fungi are involved either in dissolved N₂O uptake or N₂O production and thereby trigger stem N₂O exchange?

- We want to thank the reviewer for commenting on the possible role of mycorrhizal fungi in the soil N cycling processes and N₂O uptake / production. We have modified the discussion accordingly to account for a possible role of mycorrhizal fungi in the N turnover and N₂O exchange, and we highlighted the strong rhizospheric effects on soil N turnover processes (page 3-4, rows 120–141). Although laboratory measurements show that high N₂O soil concentrations are reflected in high stem N₂O emissions (Pihlatie et al., 2005), in an earlier study at the same site as our study, soil N₂O concentrations were reported to be rather low, close to ambient concentrations, and that production and consumption processes take place simultaneously (Pihlatie et al., 2007). Pihlatie et al., (2007) report the lowest N₂O concentration (net N₂O uptake) during spring and elevated topsoil N₂O concentration from mid-summer towards autumn.
- The roots of spruce are traditionally assumed to grow mostly close to the soil surface (e.g. Konôpka et al. 2010, Puhe 2003), whereas the roots of birch grow in both vertical and horizontal directions (Huikari 1959). Hence, it is possible that the roots of Norway spruce are mostly located in the top-most organic layer where most of the soil N₂O production and consumption also takes place (Pihlatie et al., 2007). This may be reflected in the tree stem fluxes, especially if the trees transport N₂O from the soil to the atmosphere, as supported by our data and suggested in laboratory studies (Pihlatie et al., 2005).

Line 180: You specify CO₂ burst events. Similar burst events are known for N₂O due to freezing-thawing cycles. I would have expected them to in the forest floor and tree stem emissions. To you have an explanation for the absence of such N₂O burst emissions in your experiment?

- If we assume that N₂O is mostly produced in the soil and transported via the trees to the atmosphere, we would not expect to see high burst N₂O emissions from tree stems due to the rather low N₂O production rates in the soil. We have to keep in mind, however, that we did not quantify the N cycling microbial activities in the trees, in tree wood, or on the surface of the trees and hence we cannot say anything of their potential role in the tree-N₂O-exchange.
- From previous research at the site, we know that the site is N limited, meaning that there are negligible amounts of nitrogen available for microbial N turnover processes (Pihlatie et al., 2007; Korhonen et al., 2013). Ambus et al. (2006) measured gross nitrification and denitrification rates and N₂O production pathways of forest soils within Europe, and they showed that at the Hyytiälä SMEAR II site the soil has very low nitrification activity and that most of the soil N₂O is produced via denitrification. Assuming this N limitation for the plants and microbes, the available NH₄⁺ or NO₃⁻ ions for microbial N₂O production are limited, and hence no bursts in soil N₂O production are expected. This is supported by the earlier field measurements studies (Pihlatie et al., 2007; Korhonen et al., 2013).

References

- Ambus, P., Zechmeister-Boltenstern, S., & Butterbach-Bahl, K. Relationship between nitrous oxide production and nitrogen cycling in European forests. *Biogeosciences* **3**, 135–145, 2006.
- Huikari, O. On the effect of anaerobic media upon the roots of birch, pine and spruce seedlings. *Communicationes Instituti Forestalis Fenniae* **50**,1-28, 1959.
- Konôpka, B., Moravčík, M., Pajčík, J. & Lukac, M. Effect of soil waterlogging on below-ground biomass allometric relations in Norway spruce. *Plant Biosystems* **144**, 448-457, 2010.
- Korhonen, J.F.J. et al. Nitrogen balance of a boreal Scots pine forest. *Biogeosciences* **10**, 1083–1095, 2013.
- Pihlatie, M., Ambus, P., Rinne, J., Pilegaard, K. & Vesala, T. Plant-mediated nitrous oxide emissions from beech (*Fagus sylvatica*) leaves. *New Phytol.* **168**, 93–98, 2005.
- Pihlatie, M. et al. Gas concentration driven fluxes of nitrous oxide and carbon dioxide in boreal forest soil. *Tellus B* **59**, 458–469, 2007.
- Puhe, J. Growth and development of the root system of Norway spruce (*Picea abies*) in forest stands—a review. *Forest Ecol. Manag.* **175** (1–3), 253-273, 2003.

Reviewer #3 (Remarks to the Author):

Commenting “Seasonal variation in stem N₂O exchange of boreal trees relates to their physiological activity” submitted by Dr. Katerina Machacova et al.

Boreal forests are widespread, where nitrous oxide has been largely overlooked. The authors spent a lot of time to measure annual changes in soil and stem nitrous oxide fluxes of the common boreal tree species spruce, birch, and pine. Particularly, the greenhouse gas fluxes were measured in the winter. I approve the authors worked hard and pushed this topic forward and stored useful data. However, I have the following concerns.

Stem nitrous oxide fluxes have been previously widely investigated. This work is a case investigation and lacks a novelty. When I am a reader, I cannot be impressed by the results.

- We present new and novel findings of strong seasonality in tree stem N₂O dynamics of mature trees within boreal forests. We measured tree stem N₂O fluxes from three dominant tree species in combination with a multitude of environmental and tree physiological variables. This unique data allowed us to link the N₂O fluxes to their drivers. For the first time, we also demonstrate how boreal trees act during the dormant season and how this is reflected in the annual N₂O budget.
- The mechanism of N₂O transport from the soil through the trees via transpiration stream has been demonstrated in laboratory conditions (Rusch & Rennenberg 1998; Pihlatie et al, 2005; Machacova et al. 2013). In the field, N₂O exchange of tree stems have been studied during growing seasons (Machacova et al., 2016; Machacova et al., 2017; Wen et al., 2017), whereas no other studies have captured the N₂O dynamics also during the dormant season, which can cover several months of the year in the boreal region. Accordingly, there is a lack of data and research on the seasonal variation and driving factors of the seasonality. In addition, results from laboratory studies cannot be directly applicable in the field. Trees in natural N-limited forests have not previously been expected to participate in forest ecosystem N₂O exchange because the soil N₂O production is negligible, and, assuming the only source of N₂O from tree stems is that being transported from the soil, the available N₂O to be transported is minimal. In our study, we present the first findings that trees in nutrient-poor environments take part in N₂O exchange. We further show that this N₂O exchange has a strong seasonality, which is similar, albeit different in magnitude, among the three boreal species studied.
- All these facts lead us to believe that the results presented are innovative.
- We addressed this concern of the reviewer in several parts of the manuscript by strengthening the message, as suggested also by Reviewer 1. These adjustments are visible, for example, on Page 2, rows 54–66 and Page 6, rows 237 onwards.

What is physiological activity? What are physiological parameters? Soil water content, soil and air temperatures, and photosynthetically active radiation? To my knowledge, the authors measured environmental factors but did not measure tree physiological parameters. I do not think the authors studied the physiological activity and its direct relationships with seasonal nitrous oxide fluxes.

- Tree stem CO₂ exchange, hereinafter and in the manuscript expressed as stem CO₂ efflux, is one indicator of tree physiological activity inasmuch as the net stem CO₂ efflux is a result of stem respiration, CO₂ diffusion due to water transport from the root zone to the atmosphere via the transpiration stream of the trees, and CO₂ re-fixation on the stem surface. Hence, the dynamics of stem CO₂ efflux directly reflect the respiratory activity of the tree stems and partly also the transport of water from the soil to the canopy. As a water soluble gas, CO₂ can be transported via the transpiration stream and part of the transported CO₂ then diffuses through the stem against a concentration gradient between the water in the wood and the atmosphere.
- As described above, the CO₂ efflux of tree stems is not only stem respiration but also CO₂ transported from the root zone (Aubrey & Teskey 2009; Hölttä & Kolari, 2009; Bloemen et al. 2013). Aubrey and Teskey (2009) suggest that a substantial part of below-ground autotrophic respiration (of trees) is transported from the root zone through the stems and emitted to the atmosphere from tree stems. Hölttä and Kolari (2009) show based on field data and process modelling that the stem CO₂ effluxes are comprised of both stem respiration and transport via xylem sap. Typically, CO₂ efflux measured at the bottom of tree stems, as in our study, underestimates the stem respiration due to transport of produced CO₂ through the transpiration stream, whereas CO₂ efflux measurements in the upper part of the tree stems overestimate the stem respiration due to CO₂ accumulation in the tree stems. The proportion of stem CO₂ effluxes originating from stem respiration and xylem transport cannot be generalized, however. As N₂O has good water solubility comparable to that of CO₂, we can state with confidence that xylem transport is equally possible for N₂O, and hence the N₂O emitted from tree stems can originate from internal N₂O production and diffusion of N₂O dissolved in the transpiration stream, which has been demonstrated in the laboratory (Pihlatie et al., 2005). Strong seasonality drives the sap flow and transpiration rates of

trees, and hence, as demonstrated in our study, it can be expected to have a strong effect on the stem N₂O exchange.

- Gross primary productivity (GPP) and evapotranspiration are indicators of ecosystem activity. GPP is derived from net ecosystem exchange (NEE) and modelled ecosystem respiration (Reco) ($GPP = -NEE + Reco$). NEE is the ecosystem-scale net CO₂ exchange of the forest, which mainly consists of tree CO₂ uptake in photosynthesis. Reco is derived from night-time NEE measurements and hence indicates the net respiration of the forest, which consists mostly of soil respiration and up to 50% of tree and ground vegetation respiration during summer months (Kolari et al., 2009). These are measured above the forest canopy and largely indicate the activities of the trees. Hence, we can confidently use these ecosystem measures as indicators of tree physiological activity.
- We modified the paragraph stating our study objectives to define which parameters we understand to be indicators of tree physiological and ecosystem activity in our study (page 2, rows 67–83). Furthermore, we emphasize this issue throughout the manuscript, e.g. page 3, rows 115–119.

The purpose of this study is seasonal changes in stem nitrous oxide fluxes. Why are almost half of figures on carbon dioxide? I suggest that the figures on carbon dioxide can be deleted.

- We appreciate the concern of the reviewer that the focus of the manuscript should be more on the actual N₂O exchange of trees. We have carefully considered the importance of all the figures in the manuscript and we suggest a solution that moves some of the figures to the supplementary material.
- We nevertheless do regard it as critically important to show the stem CO₂ efflux dynamics to the reader, because stem CO₂ efflux was one of the two most important drivers of the stem N₂O exchange.
- We have now moved Figs 2, 8 and 10 to the supplementary material, as they show the monthly fluxes of CO₂ from stem and forest floor (Fig. 2), effects of vegetation and dormant season to the tree stem CO₂ fluxes (Fig 8), and the annual CO₂ fluxes in tree stems and in forest floor (Fig. 10).
- We added a new figure (Fig. 3) comparing N₂O versus CO₂ fluxes in stems of birch, spruce, and pine in relation to the vegetation and dormant seasons.
- We have also moved two of the figures presenting statistical path analysis (Figs. 4 and 6) to the supplementary material, as suggested by Reviewer 1.

This work did not study the mechanisms of nitrous oxide production and emissions. The authors provided too much speculative discussion without evidence. I suggest that the discussion can be largely shortened and improved.

- We agree with the reviewer that this study is not a process study conducted in controlled laboratory conditions, which would enable detailed analysis of the N₂O exchange processes. Unfortunately, mature trees cannot be taken into the laboratory, nor can the environmental conditions in the field be controlled to such extent that certain processes could be pinpointed. An additional limitation in the field in such a long-term measurement station as Hyytiälä SMEAR II is that the use of stable isotopes is forbidden in order to avoid contamination of the site for future studies. We nevertheless present a comprehensive data set of a newly identified N₂O source in boreal forests, namely N₂O fluxes from boreal trees, and we relate the N₂O fluxes to a unique combination of tree physiological and environmental variables measured simultaneously at the site. The novelty of the study consists in the fact that there are no previous measurements of seasonality in tree N₂O dynamics anywhere in the world. Another strength is that we link the field-scale tree N₂O fluxes to a series of simultaneously measured variables, which allow studying N₂O exchange process drivers and links to the physiological activity of the trees.
- As suggested, we have carefully analysed the manuscript and removed statements that we consider overly speculative. We strongly believe, however, that our findings of strong seasonality in N₂O emissions from boreal trees, as well as the links between stem N₂O fluxes and stem CO₂ efflux, evapotranspiration, and GPP are unique findings and show for the first time that tree-N₂O dynamics

at field scale are driven by the activity of the trees and the ecosystem as a whole. To highlight this novelty, and also in responding to a request of Reviewer 1, we have strengthened the main message of our study. In addition, we have improved the description of the environmental variables with respect to the indicators of tree physiological activity (stem CO₂ efflux, evapotranspiration, GPP) (e.g. page 2, rows 67–83).

References

- Aubrey, D. P. & Teskey, R. O. Root-derived CO₂ efflux via xylem stream rivals soil CO₂ efflux. *New Phytol.* **184**(1), 35–40, 2009.
- Bloemen, J., McGuire, M. A., Aubrey, D. P., Teskey, R. O. & Steppe, K. Transport of root-respired CO₂ via the transpiration stream affects aboveground carbon assimilation and CO₂ efflux in trees. *New Phytol.* **197**(2), 555–565, 2013.
- Hölttä, T. & Kolari, P. Interpretation of stem CO₂ efflux measurements. *Tree Physiol.* **29**, 1447–1456, 2009.
- Kolari, P. et al. CO₂ exchange and component CO₂ fluxes of a boreal Scots pine forest. *Boreal Environ Res.* **14**, 761–783, 2009.
- Machacova, K. et al. *Pinus sylvestris* as a missing source of nitrous oxide and methane in boreal forest. *Sci. Rep.* **6**, doi: <https://doi.org/10.1038/srep23410>, 2016.
- Machacova, K., Maier, M., Svobodova, K., Lang, F. & Urban, O. Cryptogamic stem covers may contribute to nitrous oxide consumption by mature beech trees. *Sci. Rep.* **7**, 13243, DOI: 10.1038/s41598-017-13781-7, 2017.
- Machacova, K., Papen, H., Kreuzwieser, J. & Rennenberg, H. Inundation strongly stimulates nitrous oxide emissions from stems of the upland tree *Fagus sylvatica* and the riparian tree *Alnus glutinosa*. *Plant Soil* **364**, 287–301, 2013.
- Pihlatie, M., Ambus, P., Rinne, J., Pilegaard, K. & Vesala, T. Plant-mediated nitrous oxide emissions from beech (*Fagus sylvatica*) leaves. *New Phytol.* **168**, 93–98, 2005.
- Pihlatie, M., et al. Gas concentration driven fluxes of nitrous oxide and carbon dioxide in boreal forest soil. *Tellus B* **59**, 458–469, 2007.
- Rusch, H. & Rennenberg, H. Black alder (*Alnus glutinosa* (L.) Gaertn.) trees mediate methane and nitrous oxide emission from the soil to the atmosphere. *Plant Soil* **201**, 1–7, 1998.
- Wen, Y., Corre, M.D., Rachow, C., Chen, L. & Veldkamp, E. Nitrous oxide emissions from stems of alder, beech and spruce in a temperate forest. *Plant Soil* **420**, 423–434, 2017.

Reviewers' comments:

Reviewer #2 (Remarks to the Author):

The authors have done a very good job in revising the manuscript and the quality has certainly improved a lot. I suppose that the manuscript is close to publication. The authors have considered my concern about the huge number of figures by shifting some figures to the supplementary section. Again, this also strengthens the manuscripts main message and makes it more clearly—a point raised by ref. 1 and 3.

My second point was the possible contribution of mycorrhizal fungi to the tree-derived N₂O emissions. The authors state that the discussion was modified accordingly to account for a possible role of fungi in rows 120-141. I do not agree with this. Although they discussed soil N turnover processes in the rhizosphere that can lead to N₂O production -fungi are not mentioned in this context. I understand the point that fungi are a very complex theme and certainly not a major constituent of this work. However, for the sake of completeness the authors should point out the strong connection between tree roots and fungi in the rhizosphere and the ability of fungi to produce N₂O.

The response to my last point –the absence of N₂O burst events during CO₂ burst events- was answered in the correspondence letter with regard to site-specific parameters at the research station. Moreover, respiration is a common parameter for biological activity. Increased biological activity—for example caused by an increase in temperature- goes along with an increase in respiration. This is why respiration (of CO₂ fluxes) are often presented along with other data like N₂O or CH₄ fluxes. The robust relationship between respiration and plant-derived N₂O emissions is documented in Lenhart et al. (2015) for cryptogamic covers and in Lenhart et al. (2018) for higher plants. Please add this information to the manuscript.

I think that investigating the mechanism of nitrous oxide production and emissions are far beyond the scope of this study. Investigating mechanisms of N₂O production and transport processes deserves ¹⁵N techniques in the field. I would see this as a follow-up study. However, to address this point I suggest to make it more clearly that plants are a source of N₂O itself (in the ms only Bowatte, S. et al. 2014 –as study about grass leaves-is cited, I suggest the authors refer to literature that includes a number of plant species, eg. Hakata et al. (2003)).

Lenhart K, Behrendt T, Greiner S, Steinkamp J, Well R, Giesemann A, Keppler F. 2018. Nitrous oxide effluxes from plants as a potentially important source to the atmosphere. *New Phytologist* 10.1111/nph.15455.

Lenhart K, Weber B, Elbert W, Steinkamp J, Clough T, Crutzen P, Pöschl U, Keppler F. 2015. Nitrous oxide and methane emissions from cryptogamic covers. *Global Change Biology* 21: 3889-3900.

Hakata M, Takahashi M, Zumft W, Sakamoto A, Morikawa H. 2003. Conversion of the Nitrate Nitrogen and Nitrogen Dioxide to Nitrous Oxides in Plants. *Acta Biotechnologica* 23(2-3): 249-257.

Reviewer #3 (Remarks to the Author):

I appreciate the authors carefully revised the manuscript and in detail responded to the comments. The manuscript is largely improved. To my understanding, this work is a case measurement, the authors measured fluxes and analyzed their relationships with environmental variables. Although this work is not original, it can provide data storage.

To my understanding, the authors did not measure the physiological indicators inside trees. The indicators outside trees related to physiological activity are actually environmental variables. The authors did not study the mechanisms of nitrous oxide production and emissions, and thus these environmental variables cannot be used to indicate tree's physiological activities. I can understand

that the authors might want to make this work deep/insight by using physiological activities. However, the usage of physiological activities can mislead readers.

Point-by-point response to the referees' comments

Second revised version

MS ID: NCOMMS-18-29103A

MS Title: 'Seasonal dynamics of stem N₂O exchange follow the physiological activity of boreal trees'
('Seasonal variation in stem N₂O exchange of boreal trees relates to their physiological activity', old title)

MS Authors: Katerina Machacova, Elisa Vainio, Otmar Urban & Mari Pihlatie

Reviewer #2 (Remarks to the Author):

The authors have done a very good job in revising the manuscript and the quality has certainly improved a lot. I suppose that the manuscript is close to publication. The authors have considered my concern about the huge number of figures by shifting some figures to the supplementary section. Again, this also strengthens the manuscripts main message and makes it more clearly—a point raised by ref. 1 and 3.

*My second point was the possible **contribution of mycorrhizal fungi to the tree-derived N₂O emissions**. The authors state that the discussion was modified accordingly to account for a possible role of fungi in rows 120-141. I do not agree with this. Although they discussed soil N turnover processes in the rhizosphere that can lead to N₂O production -fungi are not mentioned in this context. I understand the point that fungi are a very complex theme and certainly not a major constituent of this work. However, for the sake of completeness the authors should point out the strong connection between tree roots and fungi in the rhizosphere and the ability of fungi to produce N₂O.*

We want to thank the reviewer for all his valuable comments which definitely highlight important aspects of N₂O production and mechanisms in plants and mycorrhizal fungi. We have now carefully read through and point out these aspects in the discussion.

a) Contribution of mycorrhizal fungi to the tree-derived N₂O emissions

We have added a discussion concerning the possible direct and indirect role of mycorrhizal fungi in production of N₂O in rhizosphere and therefore also in plant uptake of N₂O and its emission into the atmosphere (page 4, rows 153-163). Detailed discussion of this complex topic would go beyond the main message of our article.

The response to my last point –the absence of N₂O burst events during CO₂ burst events- was answered in the correspondence letter with regard to site-specific parameters at the research station.

*Moreover, respiration is a common parameter for biological activity. Increased biological activity-for example caused by an increase in temperature- goes along with an increase in respiration. This is why respiration (of CO₂ fluxes) are often presented along with other data like N₂O or CH₄ fluxes. The **robust relationship between respiration and plant-derived N₂O emissions** is documented in Lenhart et al. (2015) for cryptogamic covers and in Lenhart et al. (2018) for higher plants. Please add this information to the manuscript.*

b) Robust relationship between respiration and plant-derived N₂O emissions

We added information on strong relationship between CO₂ efflux and plant-derived N₂O emissions from literature (incl. new citations, thanks the Reviewer for the references) and placed it in the context of our results (page 3, rows 118-127).

*I think that investigating the mechanism of nitrous oxide production and emissions are far beyond the scope of this study. Investigating mechanisms of N₂O production and transport processes deserves ¹⁵N techniques in the field. I would see this as a follow-up study. However, to address this point I suggest to make it more clearly that **plants are a source of N₂O itself** (in the ms only Bowatte, S. et al. 2014 –as study about grass leaves-is cited, I suggest the authors refer to literature that includes a number of plant species, eg. Hakata et al. (2003)).*

*Lenhart K, Behrendt T, Greiner S, Steinkamp J, Well R, Gieseemann A, Keppler F. 2018. Nitrous oxide effluxes from plants as a potentially important source to the atmosphere. *New Phytologist* 10.1111/nph.15455.*

*Lenhart K, Weber B, Elbert W, Steinkamp J, Clough T, Crutzen P, Pöschl U, Keppler F. 2015. Nitrous oxide and methane emissions from cryptogamic covers. *Global Change Biology* 21: 3889-3900.*

*Hakata M, Takahashi M, Zumft W, Sakamoto A, Morikawa H. 2003. Conversion of the Nitrate Nitrogen and Nitrogen Dioxide to Nitrous Oxides in Plants. *Acta Biotechnologica* 23(2-3): 249-257.*

c) Plants as own source of N₂O

In the revised version of the manuscript we have carefully highlighted the ability of plants to produce N₂O and be itself a source of N₂O (page 4, rows 164-178). We refer to diverse literature incl. various plant species and those suggested above. We mention also the “newly detected biotic pathway of N₂O production in plants with mechanisms different from known microbial or chemical processes” (Lenhart et al. 2019).

Following literature was cited:

Smart and Bloom, 2001 [10]

Yu and Chen, 2009 [12]

Goshima et al., 1999 [21]

Hakata et al., 2003 [22]

Albert et al., 2013 [23]

Lenhart et al., 2019 [25]

Prendergast-Miller et al., 2011 [36]

Bruhn et al., 2014 [40]

Reviewer #4 (Remarks to the Author):

apologies for taking so long.

*In sum, I think that the authors did a credible job of responding to reviewer #1's comments, with the exception of the first one. I haven't seen the original manuscript, but their response to the request to **simplify the message and focus on the main message** instead produced a list of 5 main messages (!). They seem to have missed the point on this one.*

If I were editing this manuscript, I'd go back to them with some specific requests for minor revisions intended to help the reader more easily extract the key message: revise the title, clarify in the abstract, and more clearly communicate important results in topic sentences of results section. I usually don't get this explicit, but in a case like this it sometimes helps.

Hope you find this perspective useful.

We would like to thank the reviewer for pointing out the aspect of the key message and its clarity improvement. Our key message is that the newly detected seasonal dynamics of stem N₂O exchange of boreal trees follow their physiological activity.

a) We revised the title of our manuscript as follows: Seasonal dynamics of stem N₂O exchange follow the physiological activity of boreal trees. We considered also the formal requirements on title with word limit of 15 words.

b) We reorganised and modified the text of the abstract to highlight our key message and to improve the clarity of the main message to readers.

c) We added a final paragraph in the introduction section summarising the major results of our study and highlighting the key message of the study (page 2-3, rows 86-92; also in accordance with formal requirements of *Nature Communications*).

d) We adapted the first subheading of the result section to make the main message more understandable and highlighted the novelty of the study findings (page 3, row 95-98). Furthermore, we added a concluding paragraph to the first part of the Result section (similar to already available summaries of other sections; page 4, rows 179-183).

e) We modified our final concluding paragraph to simplify the main message of the paper (page 8, rows 331-338).

Reviewer #3 (Remarks to the Author):

I appreciate the authors carefully revised the manuscript and in detail responded to the comments. The manuscript is largely improved. To my understanding, this work is a case measurement, the authors measured fluxes and analyzed their relationships with environmental variables. Although this work is not original, it can provide data storage.

To my understanding, the authors did not measure the physiological indicators inside trees. The indicators outside trees related to physiological activity are actually environmental variables. The authors did not study the mechanisms of nitrous oxide production and emissions, and thus these environmental variables cannot be used to indicate tree's physiological activities. I can understand that the authors might want to make this work deep/insight by using physiological activities. However, the usage of physiological activities can mislead readers.

We would like to thank the reviewer for all the comments. We pointed out in our previous responses to the reviewers' comments, why the CO₂ exchange of tree stems can actually be used as an indicator of tree physiological activity. We also state in the introduction of the manuscript that the "stem CO₂ effluxes as well as ecosystem gross primary productivity (GPP) and evapotranspiration were considered as indicators of

physiological activity” (page 2, rows 78–80). Thus, we believe that it is clearly stated in the manuscript what was measured, and to what parameters we refer to with the *physiological activity* of trees.

Comments from the editor – article templates

At the same time, when revising your paper, please consider our article templates for the main text:

- The title which must be under 15 words, with no punctuation.

The title was shortened and modified in relation to Reviewer 4’s comments.

- The abstract (less than 150 words). It should include the background and context of the work, ‘Here we show’ or an equivalent phrase, and then the major results and conclusions of the paper. It must not contain references.

We improved the clarity and structure of the abstract, also according to Reviewer 4’s comments.

- Introduction (<1000 words), which must include the background and rationale for the work. The final paragraph should be a brief summary of the major results and conclusions. The results of the current study should only be discussed in this final paragraph.

We included a final paragraph (page 2-3, rows 86-92) to introduction part summarising the major results of our study and highlighting the key message of the study.

- Results, which must be split into subheaded sections, ensuring that the subheadings are no longer than 60 characters including spaces. Subheadings should contain no punctuation.

We shortened the first subheading in the result section (page 3, row 95).

- Discussion, without subheadings. The final paragraph of the Discussion should be your concluding paragraph.

Our paper contains one section “Results” presenting our results together with their discussion. In April, we asked the editor of *Nature Communications* Dr. Eithne Tynan concerning the format requirements and Results and Discussion section. She gave us the following answer:

“Indeed our style marks separate "Results" and "Discussion" sections. However, we allow authors to just have a "Results" section and have discussion interweaved with the description of results. So we don't mandate that you have a "Discussion" section per se, but the section should be called "Results" not "Results and Discussion", and have the appropriate subheadings.”

The final paragraph of the Results is indeed our concluding paragraph (page 8, rows 331-338).

- Methods, which must be split into subheaded sections, ensuring that the subheadings are no longer than 60 characters including spaces. There is no word limit for the Methods section.

Our method section meets the above mentioned requirements.

And for the supplementary information: The Supplementary Information should contain subheaded sections. Permitted subheadings are: Supplementary Figures, Supplementary Tables, Supplementary Notes (which are numbered Supplementary Note 1, Supplementary Note 2, etc.), and Supplementary References. Please group subheadings of the same type together and the file should be ordered in the sequence mentioned above.

Our supplementary information contains only Supplementary Figures.